# Comparative single-cell and spatial profiling of anti-SSA-positive and anti-centromere-positive Sjögren's disease reveals common and distinct immune activation and fibroblast-mediated inflammation

Jun Inamo [1,2,11] ✉, Masaru Takeshita [1,11] ✉, Katsuya Suzuki[1,3], Kazuyuki Tsunoda[4], Satoshi Usuda[4], Junko Kuramoto[5], Jonathan Moody [6], Chung-Chau Hon [6,7], Yoshinari Ando [6], Takashi Sasaki [8], Kazutoshi Yoshitake[9], Susumu Mitsuyama [9], Shuichi Asakawa[9], Yae Kanai[5], Tsutomu Takeuchi[1,10] & Yuko Kaneko[1]

Sjögren's disease (SjD) is an autoimmune disease that causes salivary gland dysfunction due to immune-mediated destruction. While autoantibodies such as anti-SSA and anti-centromere (CENT) are associated with distinct clinical manifestations, the molecular features remain to be elucidated. In this study, we apply multi-modal single-cell technologies: single-cell RNA sequencing, T cell and B cell receptor sequencing and spatial transcriptomics to salivary gland lesions, aiming to elucidate common and unique cellular and transcriptional signatures linked to different autoantibody profiles. Our analysis demonstrates that $GZMB^+GNLY^+$ CD8$^+$ T cells are the main expanded subset across different autoantibody statuses, highlighting their central role in SjD pathogenesis, while the enrichment of memory B cells is more prominent in anti-CENT-positive patients. Cytokine signaling also differs by autoantibody profile, with an activated interferon signature in anti-SSA-positive patients, whereas TGFβ signaling is enhanced in anti-CENT-positive patients. Furthermore, spatial profiling reveals $THY1^+$ fibroblasts, expressing complement genes and chemokines, as key hubs orchestrating inflammation within the salivary glands. These findings deepen our understanding of the pathogenesis of SjD, and may inform the development of targeted and personalized therapeutic strategies.

Sjögren's disease (SjD) is an autoimmune disease primarily affecting the exocrine glands, characterized by lymphocytic infiltration leading to glandular dysfunction with potentially severe systemic complications, including lymphoma and interstitial lung disease[1,2]. While much progress has been made in understanding the pathophysiology of SjD, the treatment landscape for SjD remains predominantly symptomatic, underscoring the unmet need for the development of targeted therapies[3].

---

The progressive destruction of salivary glands in SjD is mediated by infiltrating immune cells. Previous studies have revealed the presence of activated helper T cells, including Th1, Th17, and Tfh/Tph subsets, as well as *GZMB*[+]*ITGAE*[+] CD8[+] T cells and *GZMK*[+] CD8[+] T cells, in the salivary glands, implicating them in glandular damage[4–9]. Clinical manifestations of patients with SjD show biological evidence of B-cell activation, including serum polyclonal hypergammaglobulinemia, positivity for autoantibodies, and increased risk of developing B-cell lymphoma[10]. Our group performed detailed antigen specificity of B cells infiltrated into the salivary glands of patients with SjD and demonstrated that these B cells proliferate and differentiate in response to disease-specific autoantigens in situ[11]. As in rheumatoid arthritis (RA)[12,13] and cancers[14,15], recent insights have shed light on the pivotal role of tissue cells, particularly fibroblasts, in disease pathology[16,17]. At sites characteristic of chronic inflammation, fibroblasts undergo significant transformations. This process supports the maintenance and activity of tertiary lymphoid structures, implicating tissue cells as critical players in the persistent inflammation in the context of SjD[18,19].

Patients with SjD exhibit a diverse autoantibody profile, with anti-Sjögren's syndrome-related antigen A (SSA; also known as Ro) and anti-La/SSB being the most common and included as part of the diagnostic criteria[20]. Anti-SSA-positive patients often exhibit systemic features: they frequently have hypergammaglobulinemia, high titers of antinuclear antibody and rheumatoid factor, leukopenia, and immune-complex mediated lesions like cutaneous vasculitic purpura[21]. This subtype also displays an elevated interferon (IFN) signature, which correlates with hypergammaglobulinemia and higher disease activity[22]. Another subset of patients with SjD—comprising approximately 4–15% of all SjD cases, depending on the centromere protein targeted—is characterized by the presence of anti-centromere antibodies (CENT). These patients typically exhibit later disease onset and clinical features overlapping with systemic sclerosis (SSc), such as Raynaud's phenomenon and sclerodactyly[2,19,23–28]. Ultrasonography of the salivary glands reveals that CENT-positive patients have more hyperechoic bands (fibrosis)[29].

While CENT-positive and SSA-positive patients exhibit distinct clinical features, the shared and differential molecular features remain incompletely understood. Current diagnostic criteria may lead to the underdiagnosis of CENT-positive SjD, often classified as seronegative SjD (Sicca). Redefinition of SjD based on autoantibody status and the molecular features could pave the way for the development of targeted therapies for SjD and improved diagnostics in autoimmune diseases, emphasizing the role of specific autoantibodies.

Here, to characterize the heterogeneity of SjD, we applied multi-model single-cell technologies, including single-cell RNA sequencing (scRNA-seq), T cell and B cell receptor (TCR and BCR) repertoire sequencing, and spatial transcriptome sequencing, to salivary glands from patients with anti-SSA-positive SjD, CENT-positive SjD, double-positive SjD, and Sicca. Ultimately, our data show that the molecular basis underlies the clinical similarities and differences between anti-SSA and CENT-positive patients with SjD.

## Results

### Multi-modal single-cell profiling reveals heterogeneous immune and tissue cell subsets in salivary glands

To investigate the molecular differences and commonalities in patients with SjD with distinct autoantibody profiles, we collected salivary gland biopsies from 19 seropositive primary patients with SjD (SSA+, $N = 8$; CENT+, $N = 7$; SSA + CENT+, $N = 4$) and 8 seronegative (Sicca) participants for scRNA-seq and TCR/BCR repertoire analysis (Fig. 1a). A detailed summary of clinical and immunological characteristics is provided in Supplementary Data 1, showing distinct patterns across autoantibody subgroups. For example, CENT+ patients tended to be older and exhibited more frequent features, such as Raynaud's phenomenon and skin sclerosis, while SSA+ patients showed higher IgG levels. After correcting background noise and technical batch (Methods, Supplementary Fig. 1), we observed cell clusters including T cells, B/plasma cells, myeloid cells, and tissue cells in scRNA-seq data (Fig. 1b, c). Our scRNA-seq data contains 94,577 cells derived from the salivary glands, a quantity comparable to or exceeding that analyzed in recent studies for SjD[8,30–32].

### Diverse T cell subsets with subgroup-specific activation signatures in SjD

Using scRNA-seq data, we found that heterogeneous T cell subsets infiltrate the salivary glands, characterized by distinct subsets of both CD8[+] and CD4[+] T cells. Among the CD8[+] T cells, we identified two main subpopulations: *GZMK*[+] and *GZMB*[+]*GNLY*[+] cells, which likely reflect different functional states (Fig. 2a, b). Specifically, *GZMB*[+]*GNLY*[+] CD8[+] T cell subsets exhibited higher expression of cytotoxic genes such as *GZMB*, *PRF1*, and *GNLY*, while *GZMK*[+] CD8[+] T cell subsets showed elevated levels of *GZMA*, *CCL4*, and *NKG7* (Supplementary Fig. 2a, b). CD4[+] T cell subsets included central memory CD4[+] T cells (CMCD4), Tph/Tfh, Th1, Th17, and Treg cells. In addition, a small number of *GZMA*[+] CD4[+] T cells, double-negative (CD4[−]CD8[−], DN) T cells, and proliferating T cells were also detected.

Analysis using TCR repertoire revealed that the *GZMB*[+]*GNLY*[+] cytotoxic CD8[+] T cell subsets were the most clonally expanded subset within the salivary glands, followed by *GZMK*[+] CD8[+] T cells (Fig. 2c). We found that TCR repertoire diversity was lower in CD8[+] T cells compared to CD4[+] T cells, indicating their proliferation in situ (Fig. 2d). This finding aligns with the observed higher clonotype connectivity within different CD8[+] T cell subsets compared to CD4[+] T cell subsets. The pattern of dominant clonal expansion in *GZMB*[+]*GNLY*[+] cytotoxic CD8[+] T cell subsets was consistently observed across all SjD subtypes (Fig. 2e).

When we compared the diversity of the TCR repertoire across autoantibody subgroups, there was a consistent trend with the highest diversity in patients with SSA + CENT+, regardless of the parameter $q$ for the weight given to frequency in clones (Fig. 2f). It was hypothesized that the diversity of the corresponding antigens may be linked to the diversity of the TCR[11,33].

We previously found distinct gene expression profiles associated with different autoantibodies in skin lesions from patients with SSc, suggesting that antibody status reflects unique molecular features in autoimmune diseases[34]. In the context of SjD, published study using blood samples identified distinct subtypes in SjD: one with a strong IFN signature and another without, which correlated with the presence of SSA antibodies[35]. Based on these observations, we hypothesized that SSA and CENT antibodies may reflect distinct molecular features within the salivary glands. To test this hypothesis, we performed differential expression analyses of CD4[+] and CD8[+] T cells across autoantibody-defined SjD subgroups compared to Sicca controls (Fig. 2g, Supplementary Data 2). Across all SjD groups, CD4[+] and CD8[+] T cells exhibited significantly upregulated expression of immune activation and tissue-residency genes such as *CXCR4, CD69, ICOS*, and *TIGIT*, with SSA+ and SSA + CENT+ groups additionally showing elevated expression of IFN-stimulated genes such as *MX1, IFI44L*, and *IFNG*. CENT + CD4[+] T cells demonstrated enhanced expression of migration- and costimulation-related genes (*CCR7, CD27, CTLA4*), while CENT + CD8[+] T cells upregulated mediators including *LAG3*. Gene set enrichment analysis (GSEA) further highlighted subgroup-specific transcriptional programs (Supplementary Data 3). For instance, in CENT + CD4[+] T cells, positive enrichment of "Signal Transduction" pathways suggested the presence of activated CD4[+] T cells.

To investigate if distinct cell subsets are characteristic of different autoantibody subgroups, we performed differential abundance analysis. We found that Tph/Tfh and Treg cells were enriched in overall SjD compared to Sicca (Fig. 2h). This was consistent when analyzing

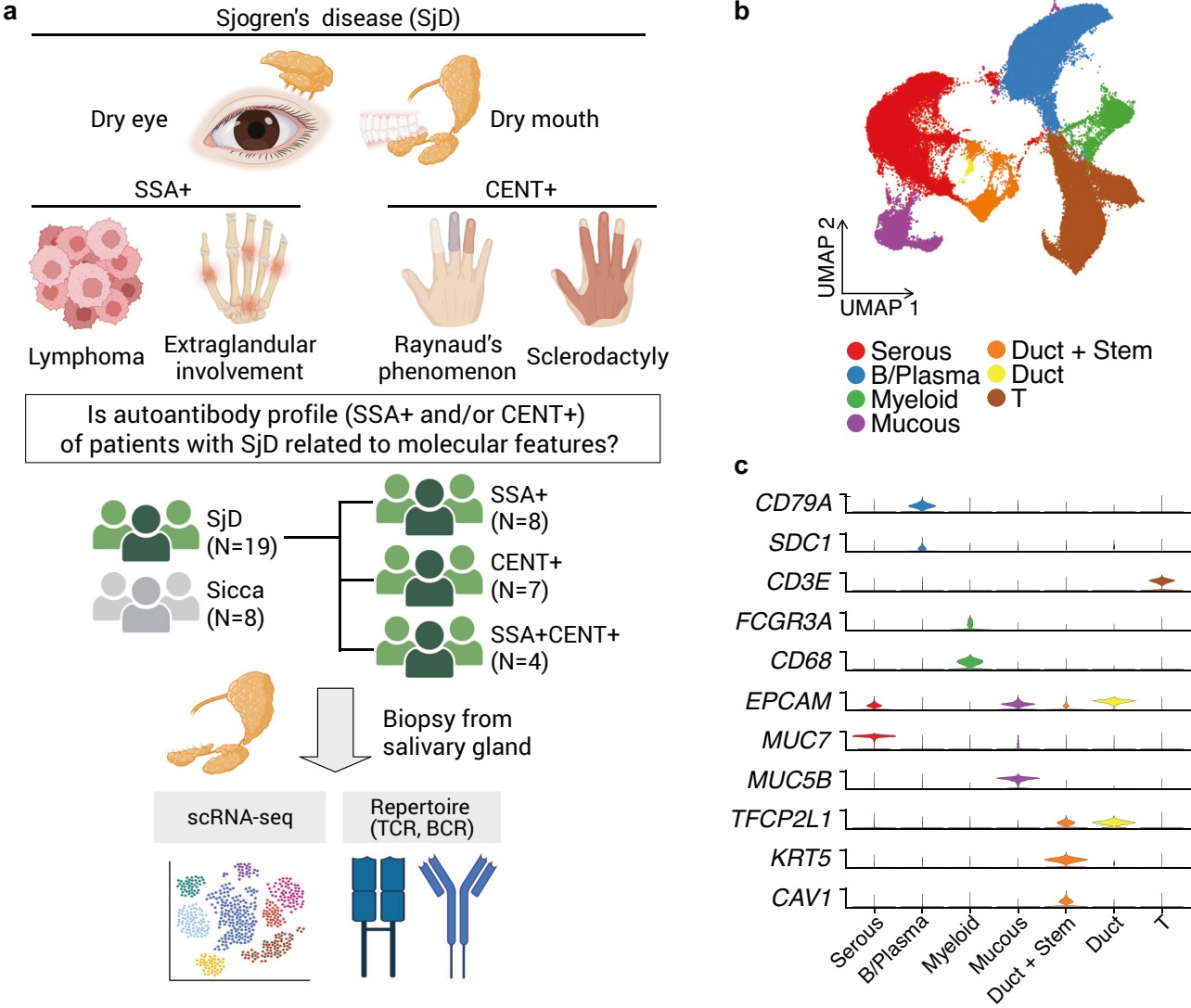

**Fig. 1 | Overview of the study design and analysis. a** Clinical overview of Sjögren's disease (SjD) and motivation for this study. The upper section illustrates common symptoms of SjD, and the lower section classifies patients based on autoantibody profiles (SSA+ and/or CENT+) by associated clinical features. The bottom part of the figure shows a schematic of the sample processing and analysis workflow, including biopsy collection from the salivary gland, followed by single-cell RNA sequencing (scRNA-seq) and T- and B-cell receptor repertoire analysis. Created in BioRender. Inamo, J. (2025) https://BioRender.com/pgjlzhs. **b** The UMAP plot illustrates broad cell types identified in the salivary glands. **c** Marker gene expressions in different cell types.

individual SjD subtypes (SSA + , CENT+, SSA + CENT+) as cases versus Sicca controls.

Although not statistically significant, we observed a skew towards more CD8[+] T cells in SSA + SjD and more CD4[+] T cells in CENT + SjD. As a sensitivity analysis, we compared SSA+ vs. CENT+ using a different algorithm and observed the congruent pattern in the skew (Supplementary Fig. 2c, d).

### Distinct B/plasma cell programs characterize CENT+ and SSA + SjD

In B/plasma cells, we observed several cell subpopulations, including plasma cells, memory B cells, and a small subset of plasmablasts (Fig. 3a, b, Supplementary Fig. 3a, b). Memory B cells expressed *ITGAX* and *TBX21*, suggesting age-associated B cells (ABCs) were included in this subset. Among plasma cells, we identified subsets with high expression of *NFKB* and *NR4A1*, associated with activation.

BCR repertoire analysis showed no obvious differences in the degree of clonal expansion between plasma cell subsets (Fig. 3c).

Memory B cells exhibited the lower diversity, suggesting proliferation within the salivary glands (Fig. 3d). Examining the relationships between BCR clonotypes across subsets revealed connectivity between different subsets. This trend was consistent when analyzing the top 20 clonally expanded clones in each disease subgroup, with the same clones spanning memory and plasma cell subsets (Fig. 3e).

Comparing BCR repertoire diversity across autoantibody subgroups revealed that the diversity differences between diseases varied depending on the weight parameter (Fig. 3f). In the situation where more weight was given to highly expanded clones ($q = 4$), we observed larger diversity in CENT + SjD than SSA + SjD and Sicca, possibly reflecting the diversity of the corresponding antigens. The higher BCR diversity in Sicca than others in scenarios where rare clones are also considered ($q = 0-1$) may be related to the higher number of singletons and other rare clones in Sicca than patients with disease-specific autoantibodies.

To further investigate disease-specific transcriptional alterations in B cells, we performed DEG analyses of memory and plasma B cell subsets across SjD subgroups compared to Sicca controls (Fig. 3g,

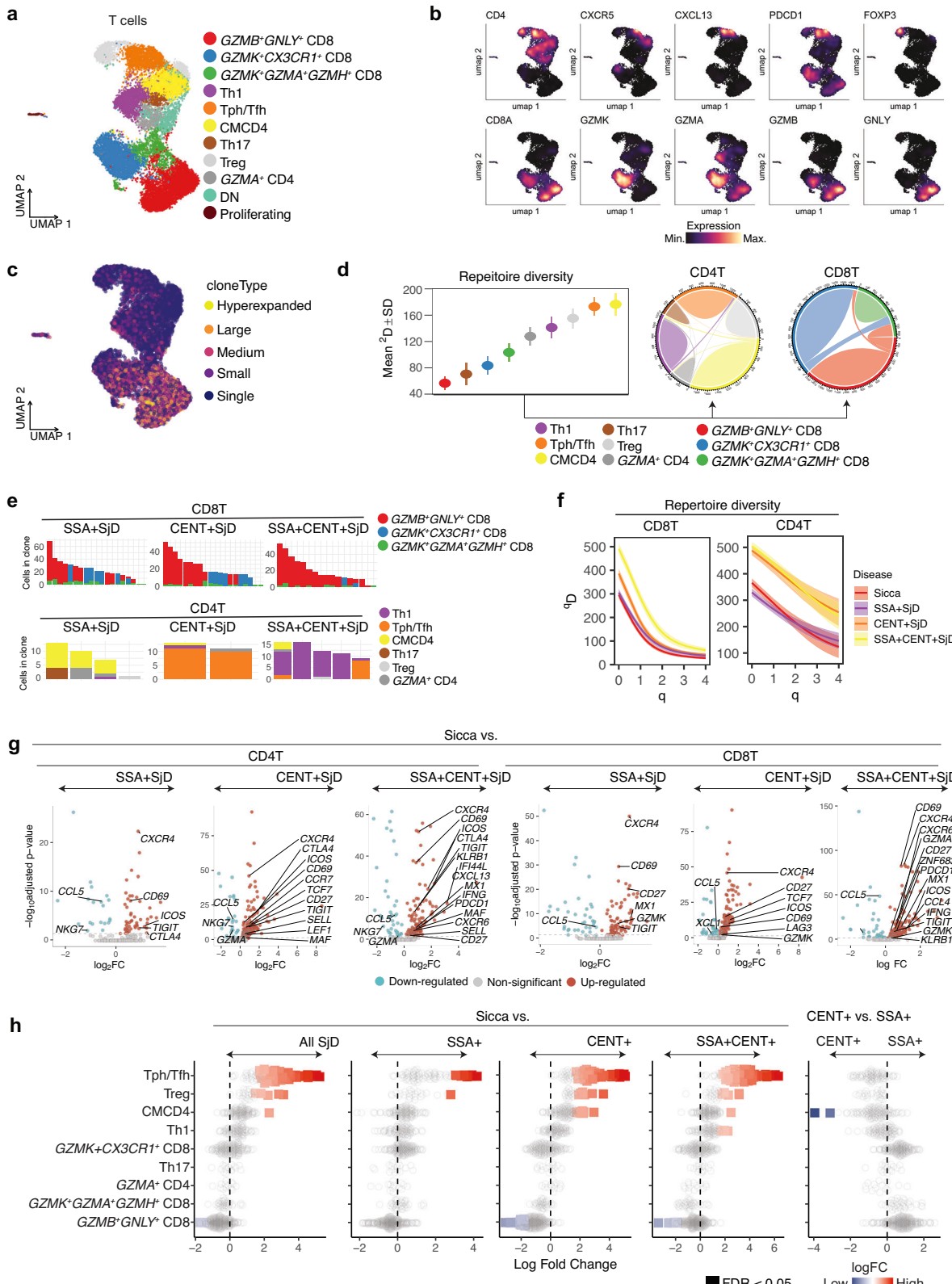

Supplementary Data 2). In memory B cells, the CENT+ and SSA + CENT+ groups showed upregulation of several genes involved in antigen presentation and survival, including *HLA-DRB1*, *CD40*, and *BCL2*. Notably, *ZEB2*, a transcription factor implicated in the differentiation toward ABCs[36], was significantly upregulated in both CENT+ and SSA + CENT+ groups, but not in SSA+ alone. These findings highlight a distinct ABC-like transcriptional signature enriched in CENT+ and double-positive patients.

GSEA comparing memory and plasma cell subsets across autoantibody subgroups revealed higher enrichment of type 1 IFN signaling pathway in plasma cell subsets of SSA+ and SSA + CENT+ patients (Supplementary Data 3).

**Fig. 2 | Cell-type-specific analysis of T cell diversity and functionality in SjD. a** The UMAP plot displays T cell subsets in salivary glands. **b** Expression levels of markers across T cell subsets. **c** Clonal expansion of T cells. **d** TCR repertoire diversity by cell clusters, with the left plot quantifying diversity scores and the right plot depicting connectivity diagrams of TCR clonotypes for different T cell subsets. Data are presented as mean values ± standard deviation (SD) derived from 200 bootstrap replicates for each group. **e** Subtype analysis of top 20 expanded clones in CD8$^+$ and CD4$^+$ T cells across different SjD autoantibody profiles. **f** TCR repertoire diversity as a smooth function ($D$) of a single parameter $q$ by autoantibody status. As the parameter $q$ increases from 0 to $+\infty$ the diversity index ($D$) depends less on rare clones and more on common (abundant) ones. Large diversity index ($D$) are interpreted as high diversity in clonal populations. Lines represent mean diversity values, and shaded areas indicate 95% confidence intervals estimated from 200 bootstrap replicates. **g** Differentially expressed gene (DEG) analysis in CD4$^+$ and CD8$^+$ T cells across SjD subgroups compared to Sicca using the Wilcoxon rank-sum test (two-sided). Volcano plots show log$_2$ fold changes (logFC) (x-axis) and $-$log$_{10}$ adjusted p-values (y-axis) for DEGs between each SjD subgroup and Sicca controls in CD4$^+$ (left three panels) and CD8$^+$ T cells (right three panels). Upregulated genes (red) and downregulated genes (blue) are highlighted with selected key genes annotated. Genes not reaching significance are shown in grey. **h** Differential abundance profiles of T cell populations in SjD subtypes. Beeswarm plots showing the distribution of logFC in neighborhoods in different cell type clusters in T cells. Each plot compares the abundance in SjD overall, SSA+, CENT+, and SSA + CENT+ subtypes versus Sicca or each other. Significant changes (false discovery rate (FDR) < 0.05) are highlighted, indicating enriched or depleted in each case. Source data are provided as a Source Data file.

In comparing analysis of cell type abundance, we observed memory B cells were increased in CENT+ subtype compared to Sicca, while plasma cells were more abundant in SSA+ group (Fig. 3h).

### Expanded B cell clones produce autoantibodies specific to disease-relevant antigens

In autoimmune diseases, particularly in SjD, autoantibody status is a crucial aspect in diagnosis. However, it remains largely unknown whether B/plasma cell clones proliferating within lesions are responsible for producing the related autoantibodies, such as SSA and CENT. To determine this, we generated antibodies from the BCR sequences of the largest clones (up to top 5 per individual) and tested their reactivity to Ro60, Ro52, and centromere antigens such as MIS12 complex and CENP-C (Fig. 3i). We observed that some proliferating BCR clones produced antibodies that specifically reacted with centromere protein and Ro60 protein, respectively, consistent with their serum reactivity. However, significant BCR clones did not target these proteins, suggesting that there may be unknown autoantibodies that react with unidentified antigens. Investigating the target of these antibodies in the future could enhance our understanding of disease pathogenesis and improve diagnostics. In Sicca samples, a single BCR clone reactive to Ro52 was identified in one patient, reflecting association of Ro52 with various autoimmune diseases and its low specificity[37–39].

### Identification of cross-phenotype tissue cell subsets in salivary glands

Next, we focused on tissue cells in salivary glands; serous acinar cells, mucous acinar cells and ducts, and stromal tissue[40]. As expected, we observed distinct subsets of tissue cells, including serous acini expressing *MUC7* and *PIP*, mucous acini expressing *CEACAM6* and *MUC5B*, and ducts expressing *KRT19* and *TFCP2L1*, and stromal cells such as endothelial and fibroblast subsets (Fig. 4a–c). Interestingly, we also identified several cell clusters with cross-phenotypes, co-expressing markers of different tissue cell subsets. For example, Duct & Stem subset co-expressed *KRT19*, a duct marker, and *KRT5* and *KRT14*, which are stem cell markers. Immunostaining also showed the presence of co-expression of duct and serous marker and duct and mucous marker, indicating the presence of transitional areas (Fig. 4d, Table 1). In contrast, the markers of the serous and mucous glands stained separately, suggesting that each gland cell differentiated independently into one or the other.

In addition to T cells and B/plasma cells, our data included other leukocyte populations in salivary glands, such as classical (CL Mono) and non-classical monocytes (NC Mono) within the myeloid compartment, as well as dendritic cells (DC) and NK cells (Fig. 4e, f, Supplementary Fig. 4a–c).

DEG analysis comparing SjD subgroups to Sicca controls revealed distinct transcriptional programs in both acinar (serous and mucous) and monocyte (classical and non-classical) populations (Fig. 4g, Supplementary Data 2). In acinar cells, SSA + SjD and SSA + CENT + SjD groups showed upregulation of IFN-stimulated genes such as *MX1* and

*ISG15*, but not in CENT+ alone. Additionally, genes related to antigen presentation (e.g., *HLA-DRB1*), stress responses (*HSPA1B, PRDX4*), and transcriptional regulation (*NR4A1, NFKBIA*) were consistently upregulated in SjD groups.

In monocytes, IFN-responsive genes such as *MX1* and *ISG15* were similarly upregulated in SSA + SjD, reinforcing a systemic IFN signature. In CENT + SjD, monocytes showed increased expression of pro-inflammatory and myeloid activation markers, including *S100A4, S100A10, FGR*, and *PRDX4*, suggestive of enhanced myeloid activation or tissue remodeling. GSEA also revealed significant enrichment of type I IFN signaling pathways in the SSA + CENT + SjD group, indicating sustained IFN-driven transcriptional activity in SSA+ subtypes (Supplementary Data 3).

In tissue cells, mucous acini were more abundant in SSA + SjD, whereas serous acini were more abundant in CENT + SjD when comparing SSA+ vs. CENT + SjD, warranting validation by immunostaining experiments using more samples (Supplementary Fig. 5).

### Correlation of clinical variables with immune and tissue cell composition across SjD subtypes

To evaluate whether cellular features are associated with clinical parameters, we performed correlation analyses between the frequency of cell clusters and clinical variables including age, disease duration, serum IgG titer, and histological focus score (Supplementary Fig. 6). We observed consistent patterns across all patients, including a negative correlation between acinar (serous and mucous) cell frequency and age, and a corresponding positive correlation between ductal epithelial cell frequency and age. These trends may reflect age-related loss of secretory function and glandular remodeling.

Among immune subsets, the proportion of Tph/Tfh and Treg cells within the T cell compartment positively correlated with focus score in the overall SjD patients, suggesting their contribution to lymphocytic infiltration in inflamed glands. Interestingly, the IgG titer was not significantly correlated with the frequency of B or plasma cells in any of the subgroups. This suggests that systemic B cell activation may reflect activity occurring outside the labial salivary gland, such as lymphoid tissues.

Subtype-specific associations were also noted. In the CENT+ subgroup, *THY1*$^+$ fibroblast and endothelial cell frequencies showed a unique positive correlation with age, suggesting that aging in CENT + SjD patients may contribute to the expansion or activation of stromal populations involved in tissue remodeling and fibrosis.

These findings collectively highlight both shared and subgroup-specific relationships between cell composition and clinical phenotypes in SjD.

### Spatial transcriptome reveals immune cell infiltration in salivary glands and distinct molecular profiles by autoantibody status

To further investigate the covariation between spatial location and infiltrated cell subsets in salivary glands, we performed spatial

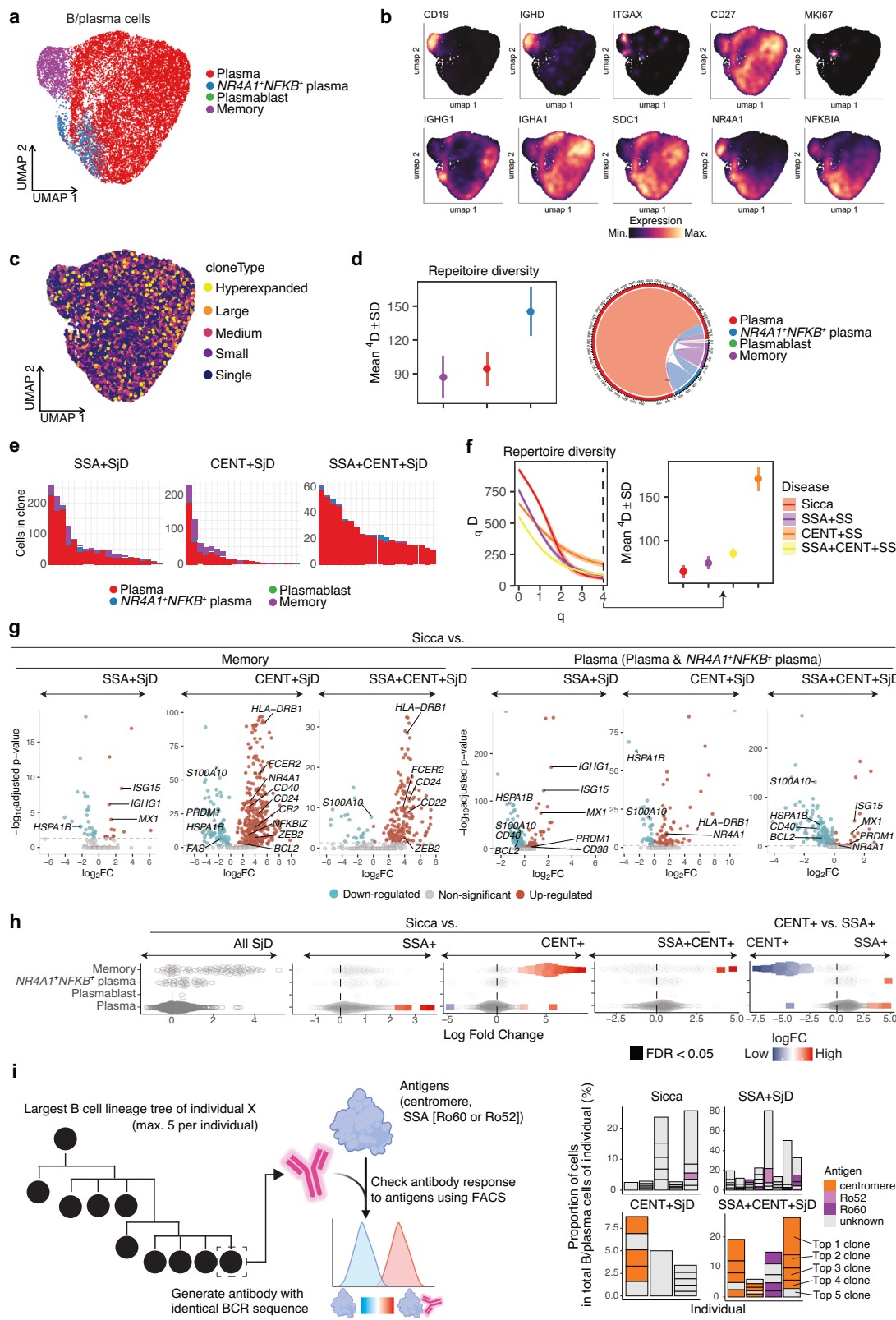

transcriptome analysis using 10X Visium on samples from patients with SjD (SSA+, $N = 6$; CENT+, $N = 5$; SSA + CENT+, $N = 4$) and Sicca ($N = 5$) (Fig. 5a, b, Supplementary Data 1). After quality control (Methods), totally 9698 spatial regions (spots) were included in downstream analysis. Cluster analysis based on transcriptome profile identified a total of 19 spatial clusters (Fig. 5c, d). Among them, cluster 4 and cluster 16 were immune-cell-rich spots (Fig. 5e).

To identify spatial clusters characteristic of SjD, we performed differential abundance analysis, as done for scRNA-seq data. The results showed that spatial cluster 4 was most enriched in SjD compared to Sicca (Fig. 5f, g). This finding was consistent across SjD subtypes (Supplementary Fig. 7). In the direct comparison between CENT+ and SSA+ patients, spatial cluster 6 was modestly but significantly expanded in CENT + SjD (Supplementary Fig. 7). This cluster was

**Fig. 3 | Comprehensive profiling of B and plasma cell subpopulations in SjD.**
**a** The UMAP displays B/plasma cell subsets in salivary glands. **b** Expression levels of markers across B/plasma cell subsets. **c** Clonal expansion of B/plasma cells. **d** BCR repertoire diversity by cell clusters, with the left plot quantifying diversity scores and the right plot depicting connectivity diagrams of BCR clonotypes for different B/plasma cell subsets. Plasmablasts were excluded from the analysis due to their low abundance. Data are presented as mean values ± standard deviation (SD) derived from 200 bootstrap replicates for each group. **e** Subtype analysis of top 20 expanded clones in B/plasma cells across different SjD autoantibody profiles. **f** BCR repertoire diversity as a smooth function (D) of a single parameter q by autoantibody status. The right plot focuses on the diversity at a specific parameter (q = 4), comparing SjD with Sicca conditions. Data are presented as mean values ± SD derived from 200 bootstrap replicates for each group. **g** Differentially expressed gene (DEG) analysis in memory and plasma cells (Plasma and $NR4A1^+NFKB^+$ plasma) across SjD subgroups compared to Sicca using the Wilcoxon rank-sum test (two-sided). Volcano plots show $\log_2$ fold changes (logFC) (x-axis) and $-\log_{10}$ adjusted p-values (y-axis) for DEGs between each SjD subgroup and Sicca controls in memory (left three panels) and plasma cells (right three panels). Upregulated genes (red) and downregulated genes (blue) are highlighted with selected key genes annotated. Genes not reaching significance are shown in grey. **h** Differential abundance profiles of B/plasma cell populations in SjD subtypes. Beeswarm plots showing the distribution of logFC in neighborhoods in different cell type clusters in B/plasma cells. Each plot compares the abundance in SjD overall, SSA+, CENT+, and SSA + CENT+ subtypes versus Sicca or each other. Significant changes (false discovery rate (FDR) < 0.05) are highlighted, indicating enriched in each case. **i** The schematic diagram on the left illustrates the strategy for investigating antibody responses to SjD-related antigens. Antibodies are generated from the largest B cell lineage trees per participant (maximum 5). Their responses to each antigen (Ro60, Ro52 and centromere) were analyzed by flow cytometry. The right bar graph shows the antigens associated with the largest trees per sample. The y-axis represents the proportion of cells in the total B/plasma cells per sample and the x-axis represents the participants analyzed. Created in BioRender. Inamo, J. (2025) https://BioRender.com/kz9vcu3. Source data are provided as a Source Data file.

composed primarily of tissue-resident stromal cells, with a minor contribution from T cells and myeloid cells. The expansion of this cluster in CENT+ patients may reflect enhanced local immune activation within the tissue microenvironment.

To determine if spatial cluster 4, which was characteristic of all SjD subtypes, exhibited subtype-specific transcriptional programs, we performed GSEA comparing each autoantibody group to the others. In CENT + SjD, TGF-β pathway, epithelial-to-mesenchymal transition (EMT) pathway, and pro-inflammatory pathways involving IL6, TNF, and type 2 IFN were upregulated (Fig. 5h). In contrast, both type 1 and 2 IFN pathways were upregulated in the SSA+ and SSA + CENT+ groups. Notably, this does not indicate the absence of IFN activation in CENT+ patients—indeed, DEG analysis comparing CENT+ to Sicca using spatial transcriptome data revealed increased expression of IFN-related genes such as *ISG15* (Supplementary Data 4). Rather, the GSEA suggests that the magnitude of IFN pathway activation is relatively higher in SSA+ patients compared to CENT+ patients.

### *THY1*+ fibroblasts are one of core players for cell-cell crosstalk in inflamed milieu

To further investigate genes associated with spatial clusters, we applied a probabilistic factor modeling approach that deconstructs spatial data in an unsupervised manner[41]. Consequently, the top latent factor, named Factor-1, was characteristic of spatial cluster 4 (Fig. 6a, b, Supplementary Fig. 8). Consistently, we observed a significant correlation between Factor-1 score and "Lymphocyte score" (based on aggregated expression level of canonical markers of T and B/plasma cells, Methods) for spatial regions (Fig. 6c), indicating that Factor-1 reflects lymphocyte infiltration in the salivary gland of SjD.

Importantly, expression of complement genes such as *C3* and *C1S*, collagen genes *COL3A1* and *COL1A1*, and chemokines *CXCL14* were correlated with Factor-1 score (Fig. 6c). To identify the cell populations expressing these top Factor-1-related genes, we examined the expression of these genes in cell subsets identified in scRNA-seq data. Importantly, all top 10 genes were expressed consistently in the *THY1*+ fibroblast cluster among the tissue cell clusters (Fig. 6e).

To further characterize the biological relevance of Factor-1, we investigated its association with spatial immune cell infiltration and gene expression. Using the Redeconve pipeline[42], with our scRNA-seq dataset from salivary glands as a reference, we deconvolved each Visium spot and estimated the proportions of T cells and B/plasma cells. Factor-1 scores were strongly correlated with T cell proportion across spatial spots (R = 0.58, p < 0.01; Supplementary Fig. 9a), whereas only a weak inverse correlation was observed with B/plasma cell proportion (R = −0.04, p < 0.01; Supplementary Fig. 9b). This discrepancy may reflect limitations in deconvolution sensitivity for B

lineage cells rather than a true absence of association. Indeed, Factor-1 values exhibited strong positive correlations with markers of B cells and plasma cells (*CD19, SDC1*), as well as T cells (*CD3E, CD4, CD8A*), cytotoxicity (*GZMK, GZMB*), and myeloid cells (*CD68*), as well as the chemokines *CXCL13* and *CCL19*, which are known to be upregulated in lymphocyte-rich regions (Supplementary Fig. 9c). In contrast, *PRG4*, the marker of a different subset of fibroblast, showed minimal correlation, underscoring the immune-specific nature of Factor-1. These findings support the interpretation of Factor-1 as a latent axis of spatial immune activation that integrates signals from multiple immune lineages and stromal components.

From these results, we hypothesized that *THY1*+ fibroblasts interact with other cell types, triggering local inflammation in the salivary glands. To investigate this, we calculated sender and receiver signals for 2186 cell-cell interaction pathways at the spot level using COMMOT[43]. Filtering in pathways that positively correlated with Factor-1 prioritized 41 pathways (Fig. 6f, Supplementary Data 5). For example, in *C3-ITGB2* interaction, both sender and receiver genes were strongly expressed in locations with high Factor-1 score (Fig. 6g). Similarly, for *CXCL14-CXCR4*, both sender and receiver genes were highly expressed in areas with high Factor-1 score (Fig. 6h). These receiver genes were expressed in subsets of T cells, B cells, monocytes, and NK cells, suggesting that the interaction between *THY1*+ fibroblasts and these immune cells may trigger local inflammation and salivary gland destruction in SjD (Fig. 6i). Immunostaining of salivary glands confirmed co-expression of CD90 (also known as THY1) and C3d in inflammatory foci (Fig. 6j).

In order to further delineate the *THY1*+ fibroblasts–immune cell interactions, suggested by our spatial transcriptomics findings, we performed a cell–cell communication analysis on our scRNA-seq data. This revealed that *THY1*+ fibroblasts exhibited a number of ligand–receptor pairs implicated *THY1*+ fibroblasts either as the signal source or target, underscoring their pivotal role in coordinating immune responses (Supplementary Fig. 10). Among these, the *C3-ITGB2* axis emerged again as an interaction signal between monocytes and *THY1*+ fibroblasts.

In this analysis, we also observed disease-specific alterations in T cell–fibroblast communication. In CENT + SjD, a broader range of cytokine and adhesion molecule interactions was detected, including *TNF–TNFRSF1A*, a pro-inflammatory signaling axis that promotes NF-κB–mediated gene expression[44]; *SIRPG–CD47*, an immune-regulatory interaction implicated IFN-γ secretion by chronically activated T cells[45]; and *CD46–JAG1*, a ligand–receptor pair that represents a crosstalk between the complement system and Notch signaling, promoting Th1 responses[46]. These interactions were absent or less pronounced in SSA + SjD. These findings suggest that global T cell activation in CENT+ patients may contribute to local

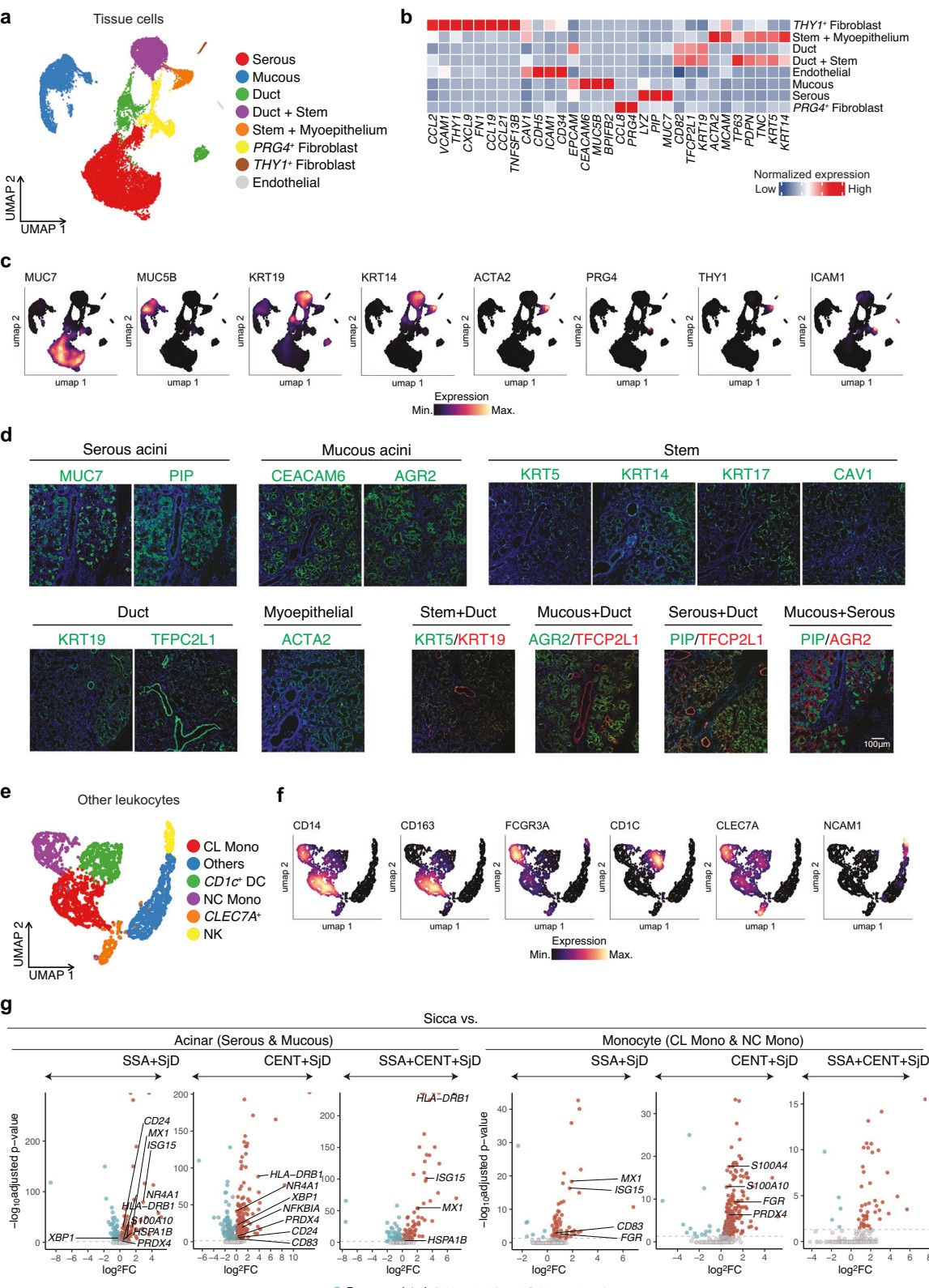

fibroblast activation, a hallmark of tissue remodeling and fibrosis, further supporting the notion of subgroup-specific pathogenic mechanisms.

Collectively, these findings highlight the complexity of cell-cell communication within the SjD salivary gland microenvironment and solidify the central role of *THY1*+ fibroblasts as potential drivers of inflammation and tissue pathology.

## Spatial transcriptomic analysis reveals shared and distinct immune-related pathways associated with *GZMK*+ and *GZMB*+ CD8+ T cells in salivary gland tissue

TCR repertoire analysis revealed two different clonal clusters of *GZMK*+ CD8+ T cells and *GZMB*+*GNLY*+ CD8+ T cells in the salivary gland. This suggests that they may proliferate and differentiate in distinct regions within the salivary gland. Previous reports have shown that *GZMK*+

**Fig. 4 | Identification of tissue cells with cross-phenotypes and identified myeloid and NK cells in salivary glands. a** The UMAP plot displaying tissue cell types. **b** Heatmap illustrates expression levels of marker genes across different tissue cell types. **c** Expression levels of markers across tissue cell subsets. **d** Immunofluorescence images displaying the localization of specific markers in various tissue structures, including serous and mucous acini, ducts, myoepithelial cells, and stem cells. The last four figures are representative of cells with cross-phenotypes of different tissue cell subsets. **e** The UMAP plot displaying subsets of myeloid cells and NK cells in salivary glands. **f** Expression levels of markers across myeloid and NK cell subsets. **g** Differentially expressed gene (DEG) analysis in acinar (Serous and Mucous) and monocytes (classical (CL Mono) and non-classical monocytes (NC Mono)) across SjD subgroups compared to Sicca. Volcano plots show $\log_2$ fold changes (logFC) (x-axis) and $-\log_{10}$ adjusted $p$-values (y-axis) for DEGs between each SjD subgroup and Sicca controls in acinar (left three panels) and monocytes (right three panels). Upregulated genes (red) and down-regulated genes (blue) are highlighted with selected key genes annotated. Genes not reaching significance are shown in grey. Source data are provided as a Source Data file.

## Table 1 | List of antibodies

| Antibody | supplier name | catalog number | clone name | Lot number | Dilution |
|---|---|---|---|---|---|
| APC anti-human CD45 Antibody | Biolegend | 304012 | HI30 | B272156 | 1/200 |
| PE anti-human CD326 (EpCAM) Antibody | Biolegend | 324205 | 9C4 | B222942 | 1/200 |
| MUC7 Antibody | NOVUS | NBP2-50391 | 4D2-1D7 | CRT/14/08 | 1/10,000 |
| PIP Antibody | NOVUS | NBP2-53226 | PIP/1571 | 5304-1P190827 | 1/400 |
| Cytokeratin 17 Antibody | NOVUS | NBP2-44427 | KRT17/778 | 3872-2P160617 | 1/500 |
| Anti-Cytokeratin 5 antibody | ABCAM | ab52635 | EP1601Y | GR3198825-2 | 1/100 |
| CEACAM6 antibody | Santa Cruz | sc-59899 | 9A6 | J1716 | 1/100 |
| AGR2 antibody | Santa Cruz | sc-101211 | 6C5 | G1019 | 1/200 |
| Cytokeratin 14 Antibody | Santa Cruz | sc-53253 | LL001 | C2218 | 1/200 |
| caveolin-1 antibody | Santa Cruz | sc-70516 | 4H312 | H2119 | 1/200 |
| Human Cytokeratin 19 Antibody | R&D Systems | MAB3506 | BA17 | XG1031809A | 1/5000 |
| LBP9 (TFCP2L1) antibody | GeneTex | GTX31477 | polyclonal | 821903659 | 1/5000 |
| Actin, Smooth Muscle Ab-1 | Thermo | MS-113-R7 | 1A4 | 113R 1208C | 1/1 |
| Goat anti-Mouse IgG1 Antibody, Alexa Fluor™ 488 | Thermo | A21131 | polyclonal | 73D1-1 | 1/500 |
| Goat anti-Mouse IgG2b Antibody, Alexa Fluor™ 488 | Thermo | A21141 | polyclonal | 1256170 | 1/500 |
| Goat anti-Rabbit IgG (H + L) Antibody, Alexa Fluor™ Plus 488 | Thermo | A32731 | polyclonal | SC243838 | 1/500 |
| Goat anti-Mouse IgG1 Antibody, Alexa Fluor™ 546 | Thermo | A21123 | polyclonal | 1249015 | 1/500 |
| Goat anti-Mouse IgG2b Antibody, Alexa Fluor™ 546 | Thermo | A21143 | polyclonal | 1711516 | 1/500 |
| F(ab')2-Goat anti-Rabbit IgG (H + L) Antibody, Alexa Fluor™ 546 | Thermo | A11071 | polyclonal | 1322319 | 1/500 |
| Polyclonal Rabbit anti-Human C3d Complement | DAKO | A0063 | polyclonal | 41602720 | 1/200 |
| Purified anti-human CD90 (Thy1) Antibody | Biolegend | 328101 | 5E10 | B349305 | 1/100 |
| Alexa Fluor® 647-F(ab')2 Goat Anti-Human IgG, Fcγ | Jackson | 109-606-170 | polyclonal | 138919 | 1/800 |
| InVivoPlus human IgG1 isotype control | BioXcell | BP0297 | polyclonal | 786120S1 | same as tested antibody (negative control) |

CD8$^+$ T cells and $GZMB^+$ CD8$^+$ T cells possess different functions. For instance, $GZMB^+$ CD8$^+$ T cells are primarily cytotoxic[47–49], whereas $GZMK^+$ CD8$^+$ T cells trigger signaling pathways associated with inflammatory conditions[50–52]. However, the impact of these functional differences at spot levels (including neighboring cells) in SjD is not yet well understood.

To investigate the respective phenotypes of spots with $GZMK^+$ CD8$^+$ T cells and $GZMB^+$ CD8$^+$ T cells, we extracted spatial regions containing CD8$^+$ T cells (678 spots in total samples). Globally overlapped, but we observed either $GZMK$ or $GZMB$ dominant spots (Fig. 7a). Within spots containing CD8$^+$ T cells, we found top correlated genes with expression of $GZMK$ and $GZMB$ including $IL32$ and $CCL5$ (Fig. 7b). Using their respective correlation coefficients, GSEA revealed common and distinct upregulated pathways that co-vary with infiltration of $GZMK^+$ and $GZMB^+$ CD8$^+$ T cells (Fig. 7c). For example, fibroblasts-related and BCR signaling pathways co-varied with both $GZMK$ and $GZMB$ expression levels (Fig. 7d). In contrast, several cytokine pathways were upregulated along with only $GZMK$ expression, while cytotoxic pathways mediated by T cells and pathways responding to IL6 were upregulated only in $GZMB^+$ spots.

## Discussion

Our multi-modal single-cell analysis of salivary gland tissues revealed distinct differences in the cellular composition in immune cell and tissue cell subsets and their transcriptional profiles by autoantibody profile.

One of the key findings was the identification of activated helper and cytotoxic T cell subsets that were abundant in patients with SSA+ and CENT + SjD. Specifically, we observed a clonal expansion of cytotoxic CD8$^+$ T cell subsets characterized by expression of $GZMB$ and $GNLY$ as a shared phenotype across SjD subtypes. This suggests that different upstream factors, depending on autoantibody status, lead to expansion of these cytotoxic T cell subsets and progressive destruction of the salivary glands. In addition, published work has shown an infiltration of $GZMK^+$ CD8$^+$ T cells in the salivary glands of patients with SjD[8], as well as our data, and they interact with epithelial cells and monocytes, contributing to inflammatory cytokine responses[52,53], suggesting that $GZMK^+$ CD8$^+$ T cells are distinct from $GZMB^+GNLY^+$ CD8$^+$ T cells and have a unique role in the mechanism of SjD.

The increased presence of helper CD4$^+$ T cells, such as Tph/Tfh and Treg, in the salivary glands of patients with SSA + SjD, CENT + SjD,

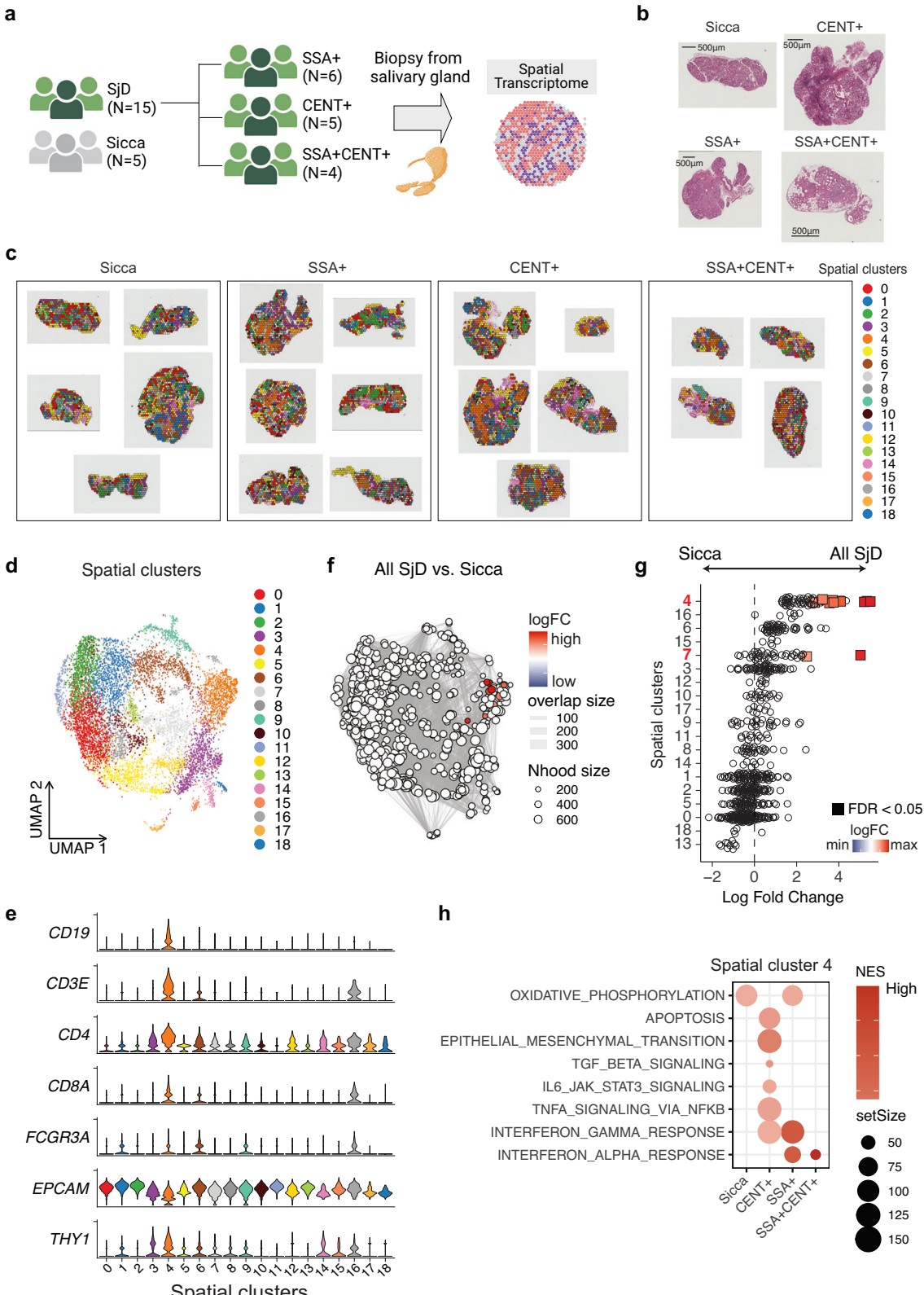

and SSA + CENT+ SjD compared to Sicca would play a crucial role in regulating the autoimmune response. These helper T cells are instrumental in modulating immune responses and could be involved in promoting or sustaining B cell activity within lesions. This enhanced B cell activity could lead to increased production of autoantibodies, as helper T cells provide necessary support for B cell differentiation and survival[54,55]. Diverse TCR repertoires of helper CD4+ T cells may result

from and contribute to the complex interplay between T cells and B cells in the glandular microenvironment, driven by chronic autoimmune activation.

Consistent with prior clinical observations, CENT-positive patients in our cohort tended to be older than CENT-negative patients (SSA+ or Sicca). This trend is particularly relevant given our transcriptional findings: CENT+ and SSA + CENT+ patients exhibited

**Fig. 5 | Spatial transcriptome analysis of salivary glands reveals enrichment of leukocyte clusters across SjD subtypes and differential pathways by auto-antibody status. a** Schema showing the study design for spatial transcriptome analysis (10X Visium). Created in BioRender. Inamo, J. (2025) https://BioRender.com/mu09iuk. **b** Hematoxylin and eosin (H&E) stained sections of representative samples used for spatial transcriptome analysis. Distribution of spatial clusters within salivary gland tissues on histological sections (**c**) and UMAP (**d**). Colors represent 19 unique spatial clusters. **e** Violin plots displaying the expression levels of key markers across the spatial clusters. **f** Neighborhood graph of spatial regions using differential abundance testing. Nodes represent neighborhoods from the

spatial clusters. Colors indicate the $\log_2$-fold change (logFC) between SjD overall and Sicca. Neighborhoods that increased in patients with SjD are shown in red. **g** Beeswarm plot showing the distribution of logFC in neighborhoods in different spatial clusters comparing the abundance between SjD overall and Sicca. Significant changes (false discovery rate (FDR) < 0.05) are highlighted, indicating enriched in all SjD. **h** Gene Set Enrichment Analysis (GSEA) comparing autoantibody status. Only selected pathways with significance (FDR < 0.05) in spot cluster 4 are shown. The size of the dots represents the set size of genes involved, and the color intensity indicates the normalized enrichment score (NES).

elevated expression of *ZEB2*, a transcription factor critical for the differentiation of ABCs, specifically within memory B cells, suggesting a distinct immunological profile associated with CENT positivity and older age. These data highlight the interplay between age, serological status, and B cell phenotypes in SjD pathogenesis, and support the hypothesis that CENT+ SjD represents a biologically distinct subset.

In our previous study[11], we found that 15.6% and 38.5% of antibody-secreting cells infiltrating the salivary glands of patients positive for serum anti-Ro60 or CENT, respectively, produced Ro60- or CENT-specific antibodies. In the present study, we further investigated the antigen specificity of B cells at the lesion site by generating recombinant antibodies from the most clonally expanded sequences within each of the top five BCR trees. Although the experimental methods differ and a direct comparison is therefore not possible, we confirmed the presence of autoreactive antibodies at a frequency consistent with previous reports.

The transcriptional differences between SSA+ and CENT+ SjD extend beyond the immune cell compartment. While the involvement of the IFN signaling pathway in the pathophysiology of SjD is well established[35,56,57], the variation in this pathway across different autoantibody-defined subgroups has not been understood. Our study highlights that distinct autoantibody profiles in SjD exhibit differential modulations in IFN signaling. Given that SSA is an RNA-binding protein (RBP), SSA-RNA complexes, such as for *Alu*, would engage TLRs on B/plasma cells, activating downstream IFN pathways[58]. In contrast, CENT+ patients exhibited activation of TGF-β pathway, EMT, and mitochondrial metabolic functions in tissue cell subsets. These pathways have been implicated in the pathogenesis of SSc[59–62], an autoimmune disease characterized by the presence of anti-CENT antibodies. In line with this finding, we previously reported that in skin lesions of SSc, the TGF-β pathway was enhanced in the CENT+ group and IFN signaling was enhanced in patients with positive antibodies to U1 ribonucleoprotein, one of the RBPs[34]. TGF-β is a key driver of fibrogenesis, playing major roles in multiple aspects of both wound healing and pathological fibrosis[63]. Targeted treatments against TGF-β have shown effectiveness in SSc[64]. Similarly, in systemic lupus erythematosus, which is also characterized by a prominent IFN signature, the expression levels of IFN signature genes have been observed to correlate positively with the efficacy of targeted therapies that inhibit the IFN pathway[65]. The delineation of differences in transcriptome signatures may aid identification of central pathways that could be targeted to treat patients with SjD with different autoantibody profiles.

The spatial localization of differentially activated pathways within the immune cell-enriched clusters highlights the importance of the local tissue microenvironment in shaping the immune response and disease progression. The co-localization of immune cells and tissue cells, such as *THY1*+ fibroblasts activated signaling pathways suggests that the interactions between infiltrating immune cells and resident tissue cells play a critical role in perpetuating the inflammatory process and driving tissue damage, as inferred in published work using scRNA-seq data[31]. Among the identified interaction pathways, we highlighted the *C3-ITGB2* and *CXCL14-CXCR4*

axes, which showed strong co-localization of sender and receiver gene expression in areas of abundant *THY1*+ fibroblasts. The *C3* complement component, expressed by *THY1*+ fibroblasts, may interact with *ITGB2* on immune cells, triggering complement-mediated inflammation[66,67]. Similarly, the chemokine *CXCL14*, secreted by *THY1*+ fibroblasts, may recruit *CXCR4*-expressing immune cells, such as T cells, memory B cells and monocytes, to the sites of inflammation[68]. Further, *CXCL14* works as an autocrine stimulator of fibroblast growth and migration[69]. The concept of fibroblasts as active participants in the pathogenesis of autoimmune diseases has gained increasing attention in recent years. Studies in RA[13,70,71], SSc[72,73], interstitial lung disease[71,74], and ulcerative colitis[71,75] have highlighted the role of fibroblasts in modulating immune cell recruitment, activation, and survival through the production of cytokines, chemokines, and extracellular matrix components. Our findings would extend this concept to SjD using multi-modal single-cell data and spatial profiling of salivary glands.

In this study, we investigated the functional heterogeneity of CD8+ T cells in the context of salivary gland inflammation using spatial transcriptomics. Our findings suggest that *GZMK*+ CD8+ T cells and *GZMB*+ CD8+ T cells may have distinct functions in the inflammatory microenvironment of the salivary gland. These results are consistent with a recent study that investigated CD8+ T cell subsets in RA synovium[51], showing that *GZMK*+ CD8+ T cells, which they termed CD8 T$_{teK}$ cells, were the predominant tissue-resident effector CD8+ T cell population in inflamed synovium and exhibited high cytokine production but low cytotoxic potential compared to *GZMB* + CD8+ T cells. Another recent work has shown that synovial fibroblasts are the major source of complement, including C3, in the inflamed synovium of RA and that *GZMK*+ CD8+ T cells activate the complement pathway by cleaving complement components C4 and C2[76]. Future studies using single-cell resolution spatial transcriptomic data and functional experiments comparing the effects of *GZMK*+ and *GZMB*+ CD8+ T cells on neighboring cells may provide further insights into their shared and unique functions in the context of SjD.

While our study provides insights into the cellular and molecular landscape of SjD salivary glands, it is important to acknowledge its limitations. One limitation is the relatively small sample size for each autoantibody subgroup. This was related to our aim in this study to elucidate detailed molecular features along with clinical categories using multi-omics layers, which inevitably limited the number of samples we could analyze. Another drawback is that, while our study identified key cell-cell interactions and signaling pathways associated with the inflammatory process in salivary glands, the functional consequences remain to be elucidated. In vitro and in vivo studies are needed to validate the role of the identified pathways in driving inflammation and tissue damage.

Finally, the approaches and findings of our study may have broader implications for the understanding and treatment of other autoimmune diseases. The identification of common pathways or cell types across different autoantibodies may provide opportunities for the repurposing of existing therapies or the development of targeted treatments.

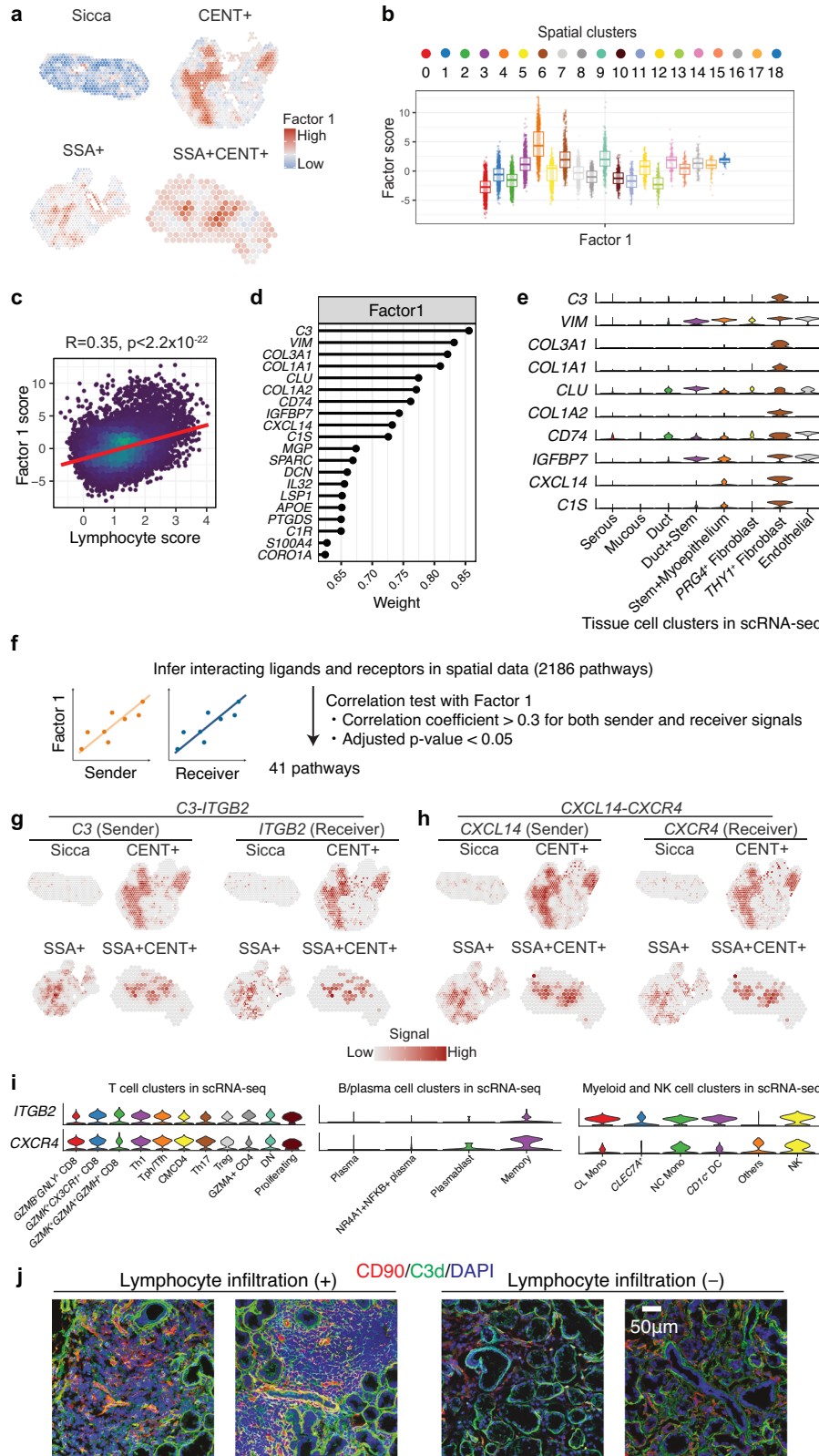

## Methods

### Participant recruitment and clinical data collection

Labial salivary glands were collected from patients who were suspected of having SjD and who underwent salivary gland biopsy for diagnosis at Keio University Hospital from October 2016 to October 2022. Clinical parameters, including immunoglobulin (Ig) and serum autoantibodies were obtained by clinical laboratory methods. IgG was measured by turbidimetric immunoassay, and anti-SSA and anti-CENT antibodies were measured by chemiluminescent enzyme immunoassay. Rheumatoid factor was measured by latex agglutination turbidimetry. The clinical characteristics of the patients were retrospectively collected from their medical records. Patients were classified into four groups according to their autoantibody profile. Patients in the Sicca group were negative for anti-SSA and anti-CENT

**Fig. 6 | Unsupervised factorization analysis reveals fibroblasts as core players in cell-cell interaction in the inflammatory foci of SjD. a** Spatial maps highlighting the Factor-1 values across different SjD subtypes and Sicca. Color intensity varies from low (blue) to high (red). **b** Box plots representing the distribution of Factor-1 values across distinct spatial clusters identified in salivary glands. The center line indicates the median, the boxes represent the interquartile range (IQR; 25th to 75th percentiles), and whiskers extend to the most extreme data points within 1.5 × IQR. **c** Scatter plot illustrating the correlation between Factor-1 levels and lymphocyte scores across tissue samples. The Spearman's correlation coefficient (*R*) and *p*-value are indicated. **d** Bar graph displaying the weights of top genes contributing to Factor-1. Higher weights suggest a stronger influence on the factor. **e** Violin plots showing the expression of key genes associated with Factor-1 across different tissue cell clusters identified in scRNA-seq data. **f** Schematic representing the methodology used to infer interacting ligands and receptors in spatial data that

covary with Factor-1. The number of pathways analyzed and the criteria for significance (Spearman's correlation coefficient and adjusted p-value) are shown. Spatial map of the expression patterns of *C3* (sender) and *ITGB2* (receiver) (**g**) and *CXCL14* (sender) and *CXCR4* (receiver) **h** across SjD subtypes and Sicca, correlating these with Factor-1. The color intensity indicates the level of expression in each tissue. **i** Receiver gene expression profiles across cell subsets identified in scRNA-seq data. **j** Representative immunofluorescence images displaying the co-localization of CD90 and C3d in foci of inflammatory cells in salivary glands of four independent individuals with SjD. In (**b**) each point represents a single spot, and the number of spots is as follows: Cluster 0: *n* = 1414; Cluster 1: *n* = 1023; Cluster 2: *n* = 1018; Cluster 3: *n* = 924; Cluster 4: *n* = 853; Cluster 5: *n* = 824; Cluster 6: *n* = 714; Cluster 7: *n* = 613; Cluster 8: *n* = 458; Cluster 9: *n* = 384; Cluster 10: *n* = 338; Cluster 11: *n* = 249; Cluster 12: *n* = 229; Cluster 13: *n* = 160; Cluster 14: *n* = 136; Cluster 15: *n* = 117; Cluster 16: *n* = 114; Cluster 17: *n* = 74; Cluster 18: *n* = 56.

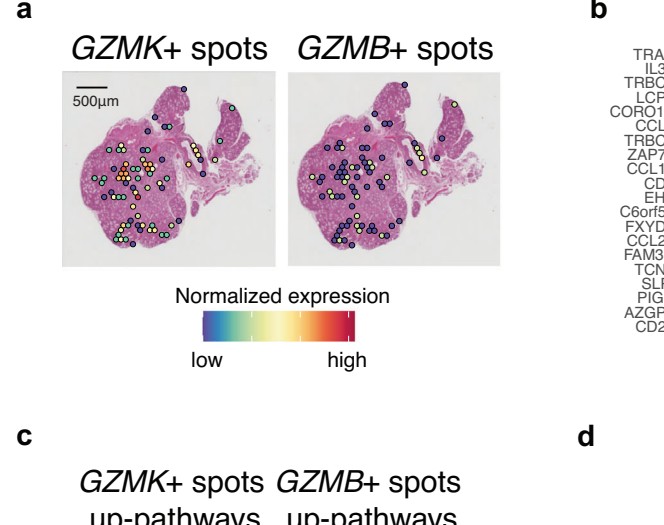

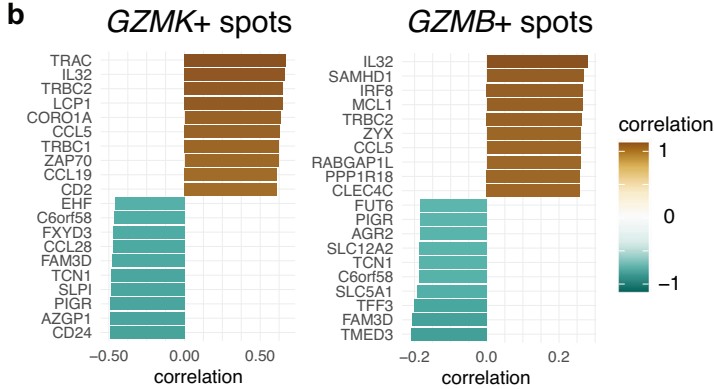

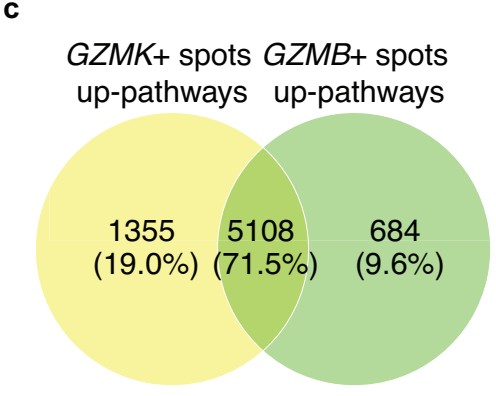

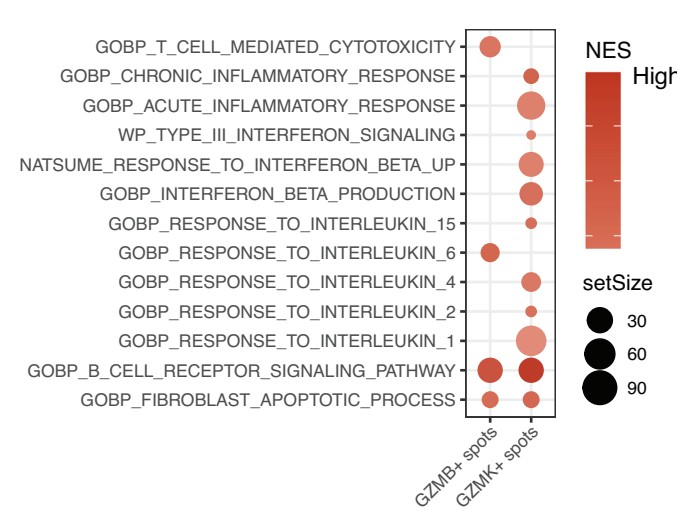

**Fig. 7 | Characterizing shared and distinct spatial transcriptome profiles associated with *GZMK*⁺ and *GZMB*⁺ in CD8⁺ T cells in salivary glands.**
**a** Expression of *GZMK* and *GZMB* in spatial regions containing CD8⁺ T cells. **b** The top 10 and bottom 10 genes correlating with *GZMK* and *GZMB* expression in spatial regions containing CD8⁺ T cells, along with their correlation coefficients by Spearman's correlation test. **c** Venn diagram displaying the up-regulated pathways

along with expression of *GZMK* and *GZMB*, respectively. **d** Gene Set Enrichment Analysis (GSEA) comparing *GZMK* and *GZMB* correlated genes. Only selected pathways with significance (false discovery rate (FDR) < 0.05) are shown. The size of the dots represents the set size of genes involved, and the color intensity indicates the normalized enrichment score (NES).

antibodies and did not meet the ACR/EULAR classification criteria for SjD[20] as well as the 1999 Japanese Ministry of Health primary SjD diagnostic criteria[77]. Patients in SSA+ and SSA + CENT+ group were positive for anti-SSA antibodies and both anti-SSA and anti-CENT antibodies respectively, and met the criteria for SjD. Patients in CENT+ group were positive for anti-CENT antibodies and met the criteria for SjD or met by counting anti-CENT antibodies as autoantibody positive. Except for one patient in the anti-SSA group, all patients did not receive any glucocorticoids or immunosuppressants

3 months prior to sampling. All procedures were approved by the medical ethics committee of Keio University Hospital and followed the tenets of the Declaration of Helsinki. All samples and information were collected after patients had provided written informed consent. There is no compensation for participants.

## Collection of sample and processing
Labial salivary gland tissue were mechanically and enzymatically digested in Advanced RPMI (Thermo, 12633012) containing 10 mg/ml

collagenase type 2 (Worthington, CLSS2) and 100 U/ml DNase1 (Merck, 04536282001) at 37 °C 20 min. The single cell suspension was centrifuged at $300 \times g$ for 5 min, and washed twice with a wash buffer (PBS containing 0.5% BSA and 2 mM EDTA). Three samples from the CENT+ group were cryopreserved in CELLBANKER 1 Plus (Zenogen, 11912) as a single cell suspension for several months. The samples were rapidly thawed and processed at a later date. After filtering through a 40 μm cell strainer, the cells were stained with fluorochrome-conjugated antibodies against CD45, CD326 and with 7-aminoactinomycin D (7AAD). The lymphocytes (7AAD-CD45+) and the salivary gland epithelial cells (7AAD-CD326+) were sorted using a FACSAria III flow cytometer (BD Biosciences). Gating strategy is shown in Supplementary Fig. 1.

## Single-cell library generation and sequencing

The sorted cells were mixed and used for single cell gene expression, BCR, and TCR analysis by Chromium Next GEM Single Cell 5′ Library & Gel Bead Kit v1.1 (10X Genomics, PN-1000165) according to the manufacturer's instructions. The generated libraries were checked using 2100 Bioanalyzer (Agilent) and sequenced using Hiseq X (Illumina) platform. Raw BCL files were demultiplexed using the *mkfastq* tool within CellRanger software (10X Genomics, version 5.0.1) with default parameters to generate FASTQ files. Subsequent to this, the samples were aligned to the human reference genome (GRCh38), and gene counts were quantified using the *count* tool within CellRanger software (10X Genomics, version 5.0.1). For the immune repertoire analysis, FASTQ files were processed separately, using the *vdj* tool within CellRanger to assemble sequences and identify clonotypes for both T-cell receptors (TCR) and B-cell receptors (BCR) across the samples.

## Quality control of single-cell RNA-seq data

After removing systematic background noise using CellBender (version 0.3.0)[78], gene expression matrices were loaded into R for further quality control. Using Seurat package (version 5.0.0)[79], the preprocessed data matrix was imported by *Read10X* and transformed to a Seurat object using the *CreateSeuratObject* function for each sample. The initial step involved a consistent quality control process applied to all cells collected from the study. For inclusion in further analysis, we removed cells that expressed fewer than 200 genes or contained more than 5% of their total UMIs mapping to mitochondrial genes. Following this preliminary filter, an additional step was employed to remove doublets using the scDblFinder[80]. Log-normalization was performed using *NormalizeData*, and the top 2000 most variable genes were identified by a variance stabilizing transformation (VST) method in *FindVariableFeatures*[81]. Principal component analysis was used for dimensionality reduction using the top 2000 highly variable genes. We further corrected batch effects and heterogeneity within samples simultaneously with the *HarmonyMatrix* function from the Harmony package[82]. After batch effect correction, the degree of mixing levels across batches was significantly increased compared to before correction. For accurate integration, we confirmed that the mixing levels for cell type, measured by LISI (Local Inverse Simpson's Index)[82,83], as equal to 1, reflecting a correct separation of unique cell types throughout the integrative embedding. For cell type analysis of T cells, B/plasma cells, tissue cells, and other leukocytes, we carried out the same normalization, feature selection, and scaling steps as described here.

## Cell type identification

After preprocessing steps, we constructed shared nearest neighbor graphs derived from the top 20 PCs and applied graph-based Louvain clustering[84] at various resolution levels. We selected optimized resolution values for all cells (0.1) based on manual check of expression of key markers in each cluster to gain the biological interpretations that made the most sense. To identify each major cell type, we checked the expression of key markers including *PTPRC*, *CD19*, *MS4A1*, *CD79A*, *SDC1*, *CD3E*, *FCGR3A*, *CD68*, *EPCAM*, *MUC7*, *MUC5B*, *TFCP2L1*, *KRT5*, *CAV1*, *PRG4*, *THY1*, and *VCAM1*. In addition, for T cells and B/plasma cells, we included cells co-expressing at least 1 productive TCR and 1 productive BCR, respectively, whereas, for tissue cells and other leukocytes, we excluded cells co-expressing at least 1 TCR or 1 BCR. For cell type analysis, we carried out the same graph-based Louvain clustering by applying optimized resolution values for each cell type (1.0 for T cells, 0.3 for B/plasma cells, 0.3 for tissue cells, 0.1 for other leukocytes).

## Single-cell TCR and BCR receptor profiling

Each sample was analyzed for TCR sequences using the filtered_contig_annotations.csv file generated by CellRanger. The analysis was conducted using Change-O[85] and scRepertoire[86]. First, we removed cells with multiple heavy chains. The identified TCR and BCR sequences, along with their corresponding cell barcodes, were then aligned with cell barcodes from the RNA library of the same sample. For downstream analysis, we included TCR and BCR with productive sequences. Clonal calls were performed by the combination of the nucleotide and gene sequence using "strict" in each function of the scRepertoire package.

## Repertoire diversity analysis

In order to assess the expansion of clones for TCR and BCR, we implemented scRepertorie[86]. Clonal diversity is calculated using Eq. (1), the generalized diversity index (Hill numbers)[87].

$$qD = \left( \sum_{i=1}^{S} p_i^q \right)^{\frac{1}{1-q}} \tag{1}$$

This method quantifies diversity as a smooth function (*D*) of a single parameter *q*. As the parameter *q* increases from 0 to + ∞ the diversity index (*D*) depends less on rare clones and more on common (abundant) ones, thus encompassing a range of definitions that can be visualized as a single curve. Large Hill Numbers are generally interpreted as high diversity in clonal populations.

## Differential gene expression and gene set enrichment analysis

Differential gene expression (DEG) analysis for the scRNA-seq data was performed using a Wilcoxon rank-sum test–based function (RunDEtest) in the SCP R package (v0.5.6)[88]. Only genes with a log2 fold change (log2FC) > 1 and Bonferroni adjusted *p*-value < 0.05 by were considered statistically significant. GSEA for the scRNA-seq data was then conducted using DEGs identified from comparisons with Sicca controls. For the Visium spatial transcriptomics data, DEG analysis was also performed using Sicca as the control group. However, to identify subgroup-specific pathways based on autoantibody status, GSEA was conducted using marker genes identified from comparisons between one autoantibody-defined subgroup and the remaining groups (e.g., CENT+ vs. others). GSEA was then performed. Gene sets were obtained from the MSigDB database[89].

## Immunohistochemistry staining

Immunohistochemical staining was performed according to previously published methods[33]. In brief, 4-μm sections of labial salivary gland were cut with a cryostat, fixed in acetone for 10 min, blocked with 5% BSA and 10% goat serum in PBS for 10 min at RT, and incubated with primary antibodies O/N at 4 °C. After washing, the slides were incubated with fluorochrome-conjugated secondary antibodies for 1 h

at RT. After washing, the slides were mounted with VECTASHIELD Mounting Medium with DAPI (Vector Laboratories, H-1500) visualized using a confocal fluorescence microscope (LSM 710 or LSM 980, Carl Zeiss).

## Assay for antibody response to clonally expanded B/plasma cell clones

Lineage tree analysis was performed using IgPhyML[90]. The 5 BCR trees with the largest number of cells from each individual were selected, and the BCR sequence with the highest number of cells in each tree was selected. The double strand DNA of variable regions of the selected BCRs were synthesized (Thermo, Strings DNA Fragments) and inserted into expression vectors for IgH or IgL[33] using NEBuilder HiFi DNA Assembly Master Mix (New England Biolabs, E2621X). Antibodies were produced by transient cotransfection of IgH and IgL vectors using the Expi293 Expression System (Thermo, A14635). Supernatants were collected after 5–7 days of culture, and antibodies were purified using Ab Capcher Mag 2 (Protenova, P-052-10). Purities of recombinant antibodies were determined by SDS-PAGE and CBB staining using a 12.5% Supersep precast gel and Quick CBB Plus (FUJIFILM Wako, 178-00551). The concentrations of recombinant antibodies were determined by ELISA for human IgG (Bethyl Laboratories, E80-104).

The reactivity for purified antibodies were determined by antigen-binding beads assay[33]. In brief, biotinylated SSA/Ro60 (DIARECT, 20300) or house-made recombinant autoantigens (SSA/Ro52, CENPA, CENPB, CENPC, MIS12 complex, CENP-HIKM, CENP-LN, CENP-OPQUR, and CENP-TWSX) with streptavidin-binding peptide tag and GFP were immobilized on Dynabeads M-280 streptavidin (Veritas, DB11206). After washing, the beads were incubated with 2 μg/ml antibody in a staining buffer, washed, and stained with AF647-conjugated anti-human IgGFc antibody. The reactivity of each antibody was measured as the median fluorescence intensity of APC using FACSVerse (BD), and data were analyzed using FlowJo software (BD). The cut-offs were determined as the higher of (75 percentile-25 percentile)x5 + median and twice the value of control IgG1 (BioXcell).

## Library generation and sequencing of spatial 10X Visium data in salivary glands

Spatial transcriptome analysis was performed using Visium CytAssist Spatial Gene Expression for FFPE (10x genomics, 1000444) according to the manufacturer's instructions. In brief, Visium Slides with 5 μm tissue sections on them were heated at 42 °C for 3 h and dried for up to 1 week. The slides were deparaffinized, stained with hematoxylin-eosin, and visualized using BZ-X710 (KEYENCE). Gene expression libraries were generated according to the protocol, checked with Agilent 2200 TapeStation, and sequenced using DNBSEQ-G400 (MGI Tech).

## Analyses of spatial transcriptome data

The initial processing of the sequencing data was conducted using spaceranger (10X Genomics, version 2.0.1), aligning the sequencing output with corresponding imaging data. Initially, we removed spatial regions that expressed fewer than 200 genes or contained more than 5% of their total UMIs mapping to mitochondrial genes, yielding 9698 spatial regions for downstream analysis. Normalization and variance stabilization using regularized negative binomial regression was performed using *SCTransform* function[91] in Seurat (version 5.0.0). To minimize batch effects by different slides, we applied *HarmonyMatrix* function from the Harmony package. We constructed shared nearest neighbor graphs derived from the top 20 PCs and applied graph-based Louvain clustering at various resolution levels and used optimized resolution values (1.0). yielding distinct 19 spatial clusters.

## Differential abundance analysis using scRNA-seq data and spatial transcriptome data

Differential abundance analysis of patients with SjD and Sicca was performed using miloR (version 1.10)[92]. This analysis identifies cell groups with varying abundance across conditions by modeling the cell count within neighborhoods formed around nodes in a k-nearest neighbor (KNN) graph. We first used the *buildGraph* function to construct a KNN graph using precomputed 30 principal components after adjusting for batch effects, where k was set to 30. Subsequently, *makeNhoods* function was applied to cluster cells into neighborhoods based on their links within the KNN graph. For assessing differential abundance, Milo utilized a negative binomial generalized linear model on the neighborhood counts, incorporating TMM normalization to address varying cell counts across samples. Age and treatment status were included as covariates in the model. As a sensitivity analysis, we performed differential abundance analysis using co-varying neighborhood analysis (can)[93] for comparison between CENT+ and SSA+. CNA employs graph-based random walks to produce a neighborhood abundance matrix (NAM), which captures the relative abundance of each sample in specific cellular neighborhoods. Differentially abundant cell populations are identified through statistical analysis of the NAM under varying conditions.

## Analysis using spatial transcriptome data

To identify latent factors associated with inflammatory spatial clusters in spatial data, we applied MEFISTO[41] with the default setting, and the number of latent factors was fine-tuned by checking the variance explained by clinical features, including autoantibody status. MEFISTO is an extension of the MOFA framework[94], which is a probabilistic factor model designed to identify principal axes of variation from data sets. MEFISTO allows for explicit modeling of spatial dependencies between samples when these dependencies are provided as covariates during model fitting. To check correlation between identified factor values and degree of infiltration of lymphocytes, we calculated "Lymphocyte score" for spatial regions by *AddModuleScore* function in Seurat using expression values of *CD3E*, *CD4*, *CD8A*, *CD19*, *MS4A1*, *IGHG1*, *IGHA1*, *SDC1*, *XBP1*, and *MZB1*.

To analyze ligand-receptor interactions in the inflammatory environment of salivary glands, we utilized the COMMOT algorithm[43] alongside the CellChat database[95,96], which includes a comprehensive list of ligand-receptor pairs. Our analysis focused on interactions between spatial regions located within 500 μm of each other. We simplified the analysis by consolidating dimeric interactions into single genes according to the CellChat database conventions. Except for these adjustments, we maintained all default settings in the COMMOT analysis.

To investigate GZMK and GZMB related pathways, CD8+ T cell containing regions were identified based on positive expression of *CD3E* and *CD8A*. To identify genes correlating with *GZMK* or *GZMB* expression, Spearman correlation coefficients were calculated between the expression of each gene and *GZMK* or *GZMB*. GSEA was performed to identify pathways co-varying with *GZMK* or *GZMB* expression separately. Gene sets were obtained from the MSigDB database (H, C2, C5, and C7 collections) using the msigdbr package[97].

## Spatial transcriptomics deconvolution analysis

To estimate cell-type proportions within each Visium spatial transcriptomics spot, we performed deconvolution using the Redeconve pipeline (v1.1.2)[42]. Our scRNA-seq dataset was used as the reference. Reference gene expression profiles were constructed by aggregating expression data across major immune and stromal cell types, including T cells, B cells, plasma cells, myeloid cells, and tissue cell subsets.

These deconvolved estimates were used to assess associations between spatial cell-type abundance and MEFISTO-inferred latent factor scores across samples.

### Cell-cell communication analysis for single-cell RNA-seq data

For cell-cell communication inference, we employed the CellChat pipeline[95,96] (v1.6.1) on single-cell RNA-sequencing data. In brief, normalized expression data and cell type cluster labels were used as input to CellChat, which integrates known ligand–receptor pairs from curated databases to predict possible intercellular interactions. Communication probability scores were generated for each ligand–receptor pair, and statistical significance was assessed using a permutation-based approach.

### Statistical and reproducibility

DEG analyses were conducted using the Wilcoxon rank-sum test (two-sided). For volcano plots, adjusted p-values were computed using the Benjamini–Hochberg method to correct for multiple testing. Correlation analyses were performed, applying Spearman's correlation with two-sided testing. For clinical variable comparisons in Supplementary Data 1, we used analysis of variance (ANOVA) for continuous variables and chi-squared tests for categorical variables. All statistical tests are noted in the corresponding figure legends and tables. A $p < 0.05$ was considered statistically significant unless otherwise specified. In vitro experiments, including an assay for antibody response were performed at least twice independently, and similar results were obtained. For representative images (e.g., micrographs, immunofluorescence), we confirmed that similar staining patterns were observed in at least two independent experiments, unless otherwise specified in the legend.

### Reporting summary

Further information on research design is available in the Nature Portfolio Reporting Summary linked to this article.

## Data availability

All raw and processed data have been deposited in the DNA Data Bank of Japan (DDBJ) via the National Bioscience Database Center (NBDC) Human Database under accession code JGAS000773 (https://humandbs.dbcls.jp/en/hum0492-v1). All other data are available in the article and its Supplementary files or from the corresponding author upon request. Source data are provided with this paper.

## Code availability

We used publicly available software for the analyses. All source code to reproduce the analyses is available on Github (https://github.com/juninamo/Keio_SjS_singlecell/) and Zenodo (https://doi.org/10.5281/zenodo.16777644).

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

## Acknowledgements

We thank Ms. Harumi Kondo, Mr. Fumitsugu Yamane, and Ms. Yukari Kaneda for collecting clinical samples and helping with the experiments. This study was supported by the Collaborative Research Resources, School of Medicine, Keio University for technical assistance. This research was performed in the Immune-mediated Inflammatory Diseases Consortium for Drug Development, supported by Keio University School of Medicine, Division of Rheumatology and Gastroenterology, Iwate Medical University, Division of Allery and Rheumatology, Department of Internal Medicine. the National Institute of Biomedical Innovation (NIBIO), Toxicogenomics Informatics Project, Ono Pharmaceutical Co., Ltd., Daiichi Sankyo Co., Ltd., Mitsubishi Tanabe Pharma CO. This publication is part of the Single Cell Medical Network of Japan. This research was also funded by Japan Society for the Promotion of Science (JSPS) Grant-in-Aids for JSPS Fellows (grant number JP21J00596) (J.I.), JSPS KAKENHI (Grant Number JP20K17430, JP20H03720, and JP22K08528) (M.T.), Keio University Academic Development Funds (M.T.), Keio Gijuku Fukuzawa Memorial Fund for the Advancement of Education and Research (MT), the Program for the Advancement of Next Generation Research Projects (Keio Univeristy) (M.T.), the Mochida Memorial Foundation for Medical and Pharmaceutical Research (M.T.), Kowa Life Science Foundation (M.T.), Japan Rheumatism Foundation (M.T.), Takeda Science Foundation (M.T.). Research funding; Ono Pharmaceutical CO., LTD., Daiichi Sankyo CO., LTD., and Mitsubishi Tanabe Pharma CO.

## Author contributions

M.T. and K.S. designed the study. M.T., K.T., and S.U. recruited patients, obtained samples, performed chromium library preparation, and curated clinical data. J.K. performed visium sample preparation and pathological evaluation. T.S. performed pre-processing of scRNA-seq data. J.I. led the computational and statistical analyses with support from J.M. C.-C.H., and K.Y. For result interpretation, J.I. and M.T. provided disease immunology inputs. J.I. wrote the manuscript with support from M.T., K.S., Y.A., S.M., S.A., Y.Kanai, T.T., and Y.Kaneko supervised the study. All authors participated in editing the final manuscript.

## Competing interests

The authors declare no competing interests.

## Additional information

[1]Division of Rheumatology, Department of Internal Medicine, Keio University School of Medicine, Shinjuku-ku, Tokyo, Japan. [2]Division of Rheumatology, University of Colorado School of Medicine, Aurora, CO, USA. [3]Division of Rheumatology, Department of Internal Medicine, NHO Tokyo Medical Center, Meguro-ku, Tokyo, Japan. [4]Department of Dentistry and Oral Surgery, Keio University School of Medicine, Shinjuku-ku, Tokyo, Japan. [5]Department of Pathology, Keio University School of Medicine, Shinjuku-ku, Tokyo, Japan. [6]Laboratory for Regulatory Genomics, RIKEN Center for Integrative Medical Sciences, Yokohama, Kanagawa, Japan. [7]Graduate School of Integrated Sciences for Life, Hiroshima University, Higashi-Hiroshima, Hiroshima, Japan. [8]Center for Supercentenarian Medical Research, Keio University School of Medicine, Shinjuku-ku, Tokyo, Japan. [9]Laboratory of Aquatic Molecular Biology and Biotechnology, Graduate School of Agricultural and Life Sciences, University of Tokyo, Bunkyo-ku, Tokyo, Japan. [10]Saitama Medical University, Iruma, Saitama, Japan. [11]These authors contributed equally: Jun Inamo, Masaru Takeshita. ✉e-mail: juninamo@keio.jp; takeshita.a5@keio.jp

