## [Transparent Peer Review file · Nature Communications]

Comparative single-cell and spatial profiling of anti-SSA-positive and anti-centromere-positive Sjögren's disease reveals common and distinct immune activation and fibroblast-mediated inflammation

Corresponding Author: Dr Jun Inamo

A version of this paper was originally rejected for publication by Nature Communications, however that decision was reconsidered after appeal by the authors.

Version 1:

Reviewer comments:

Reviewer #1

(Remarks to the Author)

The study by Jun Inamo et al. provides a comprehensive analysis of the molecular features of salivary glands in Sjögren's disease (SD) patients, categorized by anti-centromere and anti-SS-A/Ro antibody positivity, as well as the overlap of the two. Anti-centromere antibody-positive Sjögren's disease represents a distinct subgroup with clinical and immunological differences compared to classic SD patients. These differences include a higher prevalence of Raynaud's phenomenon, sclerodactyly, and autoimmune thyroiditis, alongside a lower prevalence of anti-SSA/Ro and anti-SSB/La antibodies in anti-centromere-positive patients.

The salivary gland analysis is particularly relevant, as anti-centromere-positive patients are often anti-Ro-negative and rely on biopsy for diagnostic confirmation. Thus, the study's focus on salivary gland tissue analysis is especially valuable. While the authors employ cutting-edge single-cell and spatial transcriptomic approaches, the molecular features identified across groups remain limited. Notably, the main differences are observed between all SD glands and Sicca rather than among SD subgroups, warranting further exploration.

I recommend the publication of the manuscripts, contingent upon addressing the following points.

Major Comments:

1. Title Revision

The current title is too broad and does not clearly reflect the study's focus on anti-centromere- and anti-Ro-positive subgroups in Sjögren's disease. A more precise title emphasizing these subgroups and the identified molecular features is recommended.

2. Manuscript Clarity

o The introduction should focus more on existing literature about the features of SD subgroups with different autoantibody profiles.

o The authors instead of claiming the identification of "underlying mechanisms," they should consistently use the term "molecular features," which better aligns with the data presented.

3. Detailed Summary of Clinical and Immunological Features

o The manuscript lacks a comprehensive summary of the clinical and immunological characteristics of the patient groups.

o Provide a table summarizing key features such as anti-La, RF, ANA, immunoglobulin levels, C3, C4, ESSDAI scores, systemic manifestations, Raynaud's phenomenon prevalence, treatment status, and histological parameters (e.g., focus score), where available. Highlight statistical differences for clinical variables between the groups.

This would help better understanding the data linked to the specific features of the patients analysed. For instance, anti-centromere-positive SD patients tend to be older than anti-centromere-negative patients. If this age difference is confirmed in the group analysed in the current study, it could support findings, such the observed prevalence of age-associated B cells and should be discussed in context.

4. Figures and Legends

o While the figures and legends are well-prepared, the visualization of statistical results needs improvement for better interpretability. For example, in Figure 2I, the red square in the graph overlapping with the red circles is unclear. Use a different colour or symbol to distinguish regions with $FDR < 0.05$ more effectively.

Minor Comments:

a) Terminology

Use "Sjögren's disease" instead of "Sjögren's syndrome" in line with the 2021 recommendation by Baer & Hammit (1). This terminology has been endorsed by American and European patient groups and is now widely adopted in the literature.

Reference:

Baer, A. N., & Hammit, K. M. (2021). Sjögren's disease, not syndrome. *Arthritis Rheumatol.*, 73, 1347–1348.

b) Autoantibody Overlap

Include a visualization or table showing the overlap of autoantibodies among the three groups (anti-SSA+, anti-centromere+, and anti-SSA+/anti-centromere+) and other autoantibodies such as ANA, RF, and anti-La.

c) Ro52 and Ro60 Reactivity

Specify whether patients in the anti-SSA+ and anti-SSA+/anti-centromere+ groups show seropositivity for Ro60, Ro52, or both. This detail would strengthen the interpretation of B cell reactivity and its relationship with autoantibody profiles.

(Remarks on code availability)

Reviewer #2

(Remarks to the Author)

Inamo and colleagues in this manuscript aim at understanding the mechanisms underlying the association of certain antibodies with Sjogren's Syndromes (SS). SS diagnosis is based on essentially two types of antibodies, anti-SSA and anti-centromere. Certain aspects of SS overlap with the Sicca syndrome and systemic scleroderma. The field is certainly in need of a better understanding of the pathogenesis beyond descriptive clinical diagnoses and still very superficial antibody tests. As a matter of fact, misdiagnosis and underdiagnosis are quite common.

As a consequence, this paper is timely in its approach and quite interesting for the amount of data provided. The authors studied samples from salivary glands obtained from three different categories of patients with SS stratified on the basis of single or double positivity to SSA and anti-centromere antibodies and a group of control constituting of Sicca syndrome patients. A combination of multi-modal single cell technologies, including single cell RNA sequencing, T cell and B receptor sequencing and special transcriptomics was employed. The authors tried to integrate the various datasets in a cogent picture. The approach is certainly laudable and leads to a very interesting data-heavy, perhaps too heavy, manuscript, which describes a litany of findings, separated by cell type. While some of these findings were previously described, the richness and scope of the data lend to the appreciation of this work as a potentially very useful reference paper. The limitations of this manuscript, some of them honestly acknowledged by the authors, are often inherent to the technologies used. For instance, TCR and BCR sequencing lend to data which can be interpreted only through speculation, as demonstrated also by the authors, by identification of clones, which are not necessarily specific for the antigens which have stratified the patient populations. Other limits relate to the spatial transcriptomic methodology which was employed (Visium), which does not have a single cell resolution and the results are therefore fuzzy. As a consequence, the manuscript message is less incisive than what it could be, with most of the interpretation of the data speculative rather than fact-driven. As also underscored by the authors, such limitations are also increased by the relatively small sample size. However, again, there is a certain value in this paper as a reference. What could be done to improve the paper with available data relates to the improvement of correlative analyses, by providing, for instance, network analyses. The interaction between stroma and immune cell is a very important finding, but its reporting remains somewhat shallow, both in the depth of the data produced and for their interpretation. Specifically, it would be much more informative to characterize functionally the interactions between fibroblasts and infiltrating immune cells, showing in much more detail what they express and providing more detailed pictures, possibly of direct mechanistic relevance. Ineed, the divide across auto-antibody profiles represents the most original component of this paper, and could be exploited a lot better, with more in-depth analysis of available data, and perhaps some limited additional mechanistic and validation studies. A final note relates to the figures, which are way too busy, landing, or better, deducting from clarity, particularly whenever images are provided.

(Remarks on code availability)

Reviewer #3

(Remarks to the Author)

The paper by Inamo J et al describes an interesting aim to understand the molecular and cellular derangements of autoantibody defined Sjogren's disease. The authors should be complimented for their approach using supplementary single cell transcriptomic techniques, clonality analysis and assessment of auto-antibody producing clones. The authors indicate that "redefinition of SjD based on autoantibody status and underlying molecular mechanisms could pave the way for development of targeted therapies". Although this could significantly help it is questionable whether the data that are presented are paving such a way. Overall, robust (statistically significant) and reliable findings between CENT+ and SSA+ patients and direct associations between the autoimmunity features and clinical features are lacking. There are several major and minor concerns:

Major concerns

Direct associations of autoimmunity and clinical features are missing. Differences in inflammatory markers may well be associated with heterogeneity of the patients selected. In this respect in the Supplementary table I the retrospectively collected clinical features are shown, showing that in SSA+ patients overall have more severe inflammation and overall are characterized by a higher generalized B cell hyperactivity. In addition, the clinical features of the patients are lacking in terms of systemic disease activity/organ involvement as captured by ESSDAI scores. Because CENT+ patients may also represent SSc patients, a different clinical entity, it is important to have in depth clinical characterization including skin scores (eg MRSS), etc. Unfortunately, this is lacking, making the direct relationship of the observed cellular and molecular features and autoimmunity unclear.

Although there is an enrichment of certain pathways in T and B cell types and tissue cells in the SSA+ and CENT+ groups this study does not show significant differences between these 2 groups in a direct comparison. Figures 2-7. Single cell data are studied at the group level using pooled analyses of single cells. It is unclear whether the molecular and cellular features of the individual donors are associated with the clinical features.

From the up-regulated T cell pathways that are shown in Fig 2G it appears that CENT+ patients have the lowest nr of pathways (unique pathways 66, 8, 308 and 101 for SICCA, CENT+, SSA+ and CENT+SSA+ patients, respectively). This indicates that overall T cells of CENT+ patients are characterized by reduced T cell activation. This is not reflected in the analysis of the T cell subsets in pathway upregulation (Figure 2G and 2H) and is a puzzling discrepancy. In this respect, one would think that eg. Tph/Tfh cells, highly activated cells shown to produce a considerable number of cytokines, to show some enrichment of these pathways, which however is not the case (Fig. 2H). The authors should also (next to T cell subsets) look at whether the net/global T cell activation is related to immunopathological features (eg activation of tissue cells, APCs, B cell activity, fibrosis, destruction). This could pave the way to better understanding the immune-driven differences between the groups.

Related to the previous concern, it is helpful to show the VENN diagrams for B cells, tissue cells, APCs (figure 3 and 4). This would help to appreciate the total nr of pathways that are associated with T cell dysregulation

Abundance of non-Ro-recognizing clones in SSA+ patients: It is surprising that in SSA+ patients, the most abundant clones do not seem to recognize Ro, which one would expect to define this population. This is particularly striking for patients 5, 7, and 8, where these non-Ro-recognizing clones are highly abundant. I could not find a discussion addressing this point in the manuscript. Do the authors have any data or hypotheses on what these clones might recognize? In addition, why does only 1 out of 3 (33%) CENT+ patients and 3 out of 4 CENT+SSA+ patients (75%) show CENT+ B cell clones. In this respect the authors should study if this is not just a reflection of generalized B cell hyperactivity that is often associated with SSA+ patients. In fact studying the relationship between T cell derangements and generalized B cell activity could reveal important immunopathological mechanisms.

Why did the authors not compare CENT+ and SSA+ patients in Figure 4 and switch from this comparison of patients to sicca vs all pSS patients in Figure 5? To link all the observed molecular and cellular changes it would be very valuable/necessary to have the same comparisons.

In Figure 6, why was the MEFISTO Factor analysis not performed in the separate patient (autoantibody) groups, since the aim is to find differences between these groups. The pooled analysis may generate inappropriate results. In this respect it would be reassuring to validate well known factors that are tightly correlated to lymphocytic infiltrates (in SSA+ patients eg. CCL19, CXCL13-associated pathways). Such factors previously were shown to yield far more stronger correlations compared what seems to be the case for the loadings of Factor 1 (figure 6C,D). In fact, what are the correlations of the identified loadings with T cells, B cells etc (eg assessed by absolute counts of deconvolution data).

Minor

Several observations come with some uncertainty because of differences in group size. Eg. In Figure 2H in SSA+ patients (n=8) GZMB+GNLY+CD8+ show an enrichment of the EBV pathway (likely type I IFN), but not in the SSA+CENT+ patients (n=4) nor in CENT+ patients (n=4). Is this not just a power issue?

Harmonization of single-cell data (Mat. & Meth., line 494): The manuscript mentions a harmonization step for single-cell data. Since the single-cell analysis appears to have been conducted immediately after tissue dissociation, with samples potentially processed on different days, it is unclear what specific batch effects are being addressed. Additional details on the rationale and implementation of this harmonization would be helpful.

Please use new nomenclature for Sjogren syndrome -> Sjogren's disease

(Remarks on code availability)

Reviewer #4

(Remarks to the Author)

I co-reviewed this manuscript with one of the reviewers who provided the listed reports. This is part of the Nature Communications initiative to facilitate training in peer review and to provide appropriate recognition for Early Career

Researchers who co-review manuscripts.

(Remarks on code availability)

Version 2:

Reviewer comments:

Reviewer #2

(Remarks to the Author)

The authors made sincere effort in trying to address the reviewers' comments to the best of their capability. The manuscript remains, in some cases, still hampered by the same problems which were identified previously. Altogether, however, most of the responses are satisfactory and the manuscript has substantially improved.

(Remarks on code availability)

Reviewer #3

(Remarks to the Author)

The authors have given detailed answers to all the raised points. I think the added analyses and clarifications have significantly improved the manuscript.

There is one point the authors should address:

According to the ACR/EULAR criteria to classify as a SjD the patients should have objectified dryness symptoms and either an LFS of 1 or higher or SSA positivity. For the CENT+ group this is not clear. Only 1 is SSA+ and the LFS median values don't give insight. Maybe give means or indicate which percentage has LFS 1 or higher. If the patients should not qualify, consider them CENT sicca. Were Schirmer and UWS quantified? Please add these to the table

(Remarks on code availability)

Reviewer #4

(Remarks to the Author)

(Remarks on code availability)

Reviewers' comments:

Reviewer 1 (Remarks to the Author):

The study by Jun Inamo et al. provides a comprehensive analysis of the molecular features of salivary glands in Sjögren's disease (SD) patients, categorized by anti-centromere and anti-SSA/Ro antibody positivity, as well as the overlap of the two. Anti-centromere antibody-positive Sjögren's disease represents a distinct subgroup with clinical and immunological differences compared to classic SD patients. These differences include a higher prevalence of Raynaud's phenomenon, sclerodactyly, and autoimmune thyroiditis, alongside a lower prevalence of anti-SSA/Ro and anti-SSB/La antibodies in anti-centromere-positive patients.

The salivary gland analysis is particularly relevant, as anti-centromere-positive patients are often anti-Ro-negative and rely on biopsy for diagnostic confirmation. Thus, the study's focus on salivary gland tissue analysis is especially valuable. While the authors employ cutting-edge single-cell and spatial transcriptomic approaches, the molecular features identified across groups remain limited. Notably, the main differences are observed between all SD glands and Sicca rather than among SD subgroups, warranting further exploration.

I recommend the publication of the manuscripts, contingent upon addressing the following points.

We appreciate the efforts and positive feedback relative to our revised manuscript.

Major Comments:

Reviewer 1 Comment #1

- Title Revision:
 - The current title is too broad and does not clearly reflect the study's focus on anti-centromere- and anti-Ro-positive subgroups in Sjögren's disease. A more precise title emphasizing these subgroups and the identified molecular features is recommended.

We appreciate the helpful feedback regarding our manuscript title. In accordance with the reviewer's suggestion, we have revised the title to: "**Comparative single-cell and spatial profiling of anti-SSA- and anti-centromere-positive Sjögren's disease reveals common and distinct immune activation and fibroblast-mediated inflammation**". This change more precisely reflects our focus on the distinct autoantibody subgroups and the specific cellular findings that differentiate them. We hope this revised title addresses your concern and more accurately conveys the scope and significance of our study.

Reviewer 1 Comment #2

- Manuscript Clarity:
 - The introduction should focus more on existing literature about the features of SD subgroups with different autoantibody profiles.

We agree that a more focused discussion on existing literature regarding SjD subgroups defined by different autoantibody profiles will strengthen the Introduction. We have revised the Introduction to incorporate more detailed information on the distinct clinical features associated with CENT+ SjD and SSA+ SjD.

- The authors instead of claiming the identification of "underlying mechanisms," they should consistently use the term "molecular features," which better aligns with the data presented.

We appreciate the reviewer's suggestion to describe our study more appropriately. As suggested by the reviewer, we have revised the manuscript by replacing references to "underlying mechanisms" with "molecular features" throughout to ensure that the terminology more accurately reflects our data.

Reviewer 1 Comment #3

- Detailed Summary of Clinical and Immunological Features:
 - The manuscript lacks a comprehensive summary of the clinical and immunological characteristics of the patient groups. Provide a table summarizing key features such as anti-La, RF, ANA, immunoglobulin levels, C3, C4, ESSDAI scores, systemic manifestations, Raynaud's phenomenon prevalence, treatment status, and histological parameters (e.g., focus score), where available. Highlight statistical differences for clinical variables between the groups. This would help better understanding the data linked to the specific features of the patients analysed. For instance, anti-centromere-positive SD patients tend to be older than anti-centromere-negative patients. If this age difference is confirmed in the group analysed in the current study, it could support findings, such the observed prevalence of age-associated B cells and should be discussed in context.

We appreciate the reviewer's valuable feedback.

In response to the reviewer's comment regarding the need for a comprehensive summary of patient characteristics, we have **expanded the anti-centromere(CENT)-positive SjD group from 4 to 7 samples in our scRNA-seq dataset and accordingly updated Supplementary Table 1**, which now details the clinical and immunological features of all patient groups in our study.

This table provides a thorough overview of key parameters. For instance, as noted in the reviewer's comment, anti-CENT-positive patients in our cohort tended to be older than anti-CENT-negative patients (Sicca and SSA+ SjD), although this difference was not statistically significant.

Supplementary Table 1a. Characteristics of subjects for paired scRNA-seq and repertoire data.

	Sicca	SSA+ SjD	CENT+ SjD	SSA+CENT+ SjD	p-values
N	8	8	7	4	
Age, [IQR]	54.5 [44.8, 66.8]	50.0 [42.5, 77.8]	70.0 [63.0, 78.0]	62.0 [52.0, 71.8]	0.44
Female, % (N)	75.0 (6)	100.0 (8)	85.7 (1)	75.0 (3)	0.50
Duration, month [IQR]	9.5 [2.0, 15.0]	16.0 [3.8, 31.5]	12.0 [7.0, 48.0]	18.0 [11.0, 24.0]	0.62

IgG, mg/dl [IQR]	1567.0 [1320.2, 1747.5]	1333.5 [961.2, 1521.8]	1187.0 [1048.0, 1259.0]	1744.0 [1484.5, 2041.5]	0.096
Anti-Ro60 antibody positive, % (N)	0.0 (0) ^a	100.0 (8)	0.0 (0)	100.0 (4)	<0.000 1
Anti-Ro52 antibody positive, % (N)	0.0 (0) ^b	37.5 (3)	14.3 (1)	25.0 (1)	0.36
Anti-SSB antibody positive, % (N)	0.0 (0)	37.5 (3)	14.3 (1)	50.0 (2)	0.058
ANA positive, % (N)	25.0 (2)	50.0 (4)	100.0 (7)	100.0 (4)	0.0074
Rheumatoid factor positive, % (N)	37.5 (3)	37.5 (3)	14.3 (1)	50.0 (2)	0.62
C3, mg/dL [IQR]	95.5 [81.0, 108.0]	82.0 [69.0, 97.0] ^a	105.0 [96.0, 112.5] ^b	101.5 [89.8, 117.0]	0.052
C4, mg/dL [IQR]	21.5 [12.3, 25.0]	25.0 [18.0, 28.0] ^a	22.5 [20.0, 26.0] ^b	28.5 [23.3, 30.0]	0.29
ESSDAI score, index [IQR]	NA	0.5 [0.0, 6.25]	0.0 [0.0, 2.0]	0.5 [0.0, 3.3]	0.71
Focus score [IQR]	0.0 [0.0, 0.0]	0.4 [0.0, 2.5]	0.0 [0.0, 4.2]	4.4 [0.5, 6.8]	0.040
Rainaud's phenomenon positive, % [N]	0.0 (0)	12.5 (1)	42.9 (3)	50.0 (2)	0.16
Systemic manifestations, N	0	ILD 1, CV 1, arthralgia 1	ILD 1, skin sclerosis 2 ^d	arthralgia 1	
Treatment, % (N)	0.0 (0)	12.5 (1) ^e	0.0 (0)	0.0 (0)	0.48

p-values were obtained by analysis of variance for continuous value and chi-squared test for categorical value.

a: n=7

b: n=6

c: n=3

d: Two patients showed skin sclerosis limited to the fingers (mRSS = 4). One of them met the classification criteria for limited cutaneous systemic sclerosis based on the combination of Raynaud's phenomenon and anti-centromere antibody positivity, while the other did not meet the criteria due to the absence of Raynaud's.

e: prednisolone 50 mg/day, hemodialysis, intravenous cyclophosphamide 1000 mg/month, rituximab 375 mg/m² twice 3 months prior to lip biopsy, hemodialysis for thrombotic microangiopathy and cryoglobulinemic vasculitis

ILD; interstitial lung disease, CV; cryoglobulinemic vasculitis

Supplementary Table 1b. Characteristics of subjects for Visium data.

	Sicca	SSA+ SjD	CENT+ SjD	SSA+CENT+ SjD	P-values
N	5	6	5	4	
Age, [IQR]	41.0 [39.0, 48.0]	47.5 [41.8, 63.0]	59.0 [54.0, 67.0]	62.0 [52.0, 71.8]	0.15
Female, % (N)	100 (5)	83.3 (5)	100 (5)	75.0 (3)	0.49
Duration, month [IQR]	24.0 [2.0, 36.0]	36.0 [9.0, 60.0]	60.0 [24.0, 168.0]	18.0 [11.0, 24.0]	0.62
IgG, mg/dl [IQR]	1000.5 [935.8, 1177.5]	1965.5 [1509.2, 2319.0]	1226.0 [1171.0, 1253.0]	1744.0 [1484.5, 2041.5]	0.04
Ant-Ro60 antibody positive, % (N)	0.0 (0)	100.0 (6)	0.0 (0)	100.0 (4)	0.0002
Anti-Ro52 antibody positive, % (N)	0.0 (0) ^a	75.0 (3) ^a	0.0 (0) ^a	25.0 (1)	0.42
Anti-SSB antibody positive, % (N)	0.0 (0)	66.7 (4)	0.0 (0)	50.0 (2)	0.031
ANA positive, % (N)	20.0 (1)	100.0 (6)	100.0 (5)	100.0 (4)	0.0018
Rheumatoid factor positive, % (N)	0.0 (0) ^a	83.3 (5)	25.0 (1) n=4	50.0 (2)	0.057
C3, mg/dL [IQR]	94.0 [76.8, 109.0] ^a	105.5 [78.0, 143.5] ^a	84.0 [77.0, 87.0] ^b	101.5 [89.9, 117.0]	0.57
C4, mg/dL [IQR]	19.0 [16.3, 21.8] ^a	18.0 [15.8, 22.5] ^a	14.0 [14.0, 25.0] ^b	28.5 [23.3, 30.0]	0.077
ESSDAI score, index [IQR]	NA	1.0 [0.0, 4.3]	0.0 [0.0, 1.0]	0.5 [0.0, 3.3]	0.48
Focus score [IQR]	0.0 [0.0, 0.0]	2.1 [1.2, 5.7]	4.0 [2.4, 4.7]	4.4 [0.5, 6.8]	0.021
Rainaud's phenomenon positive, % [N]	0.0 (0)	0.0 (0)	40.0 (2)	50.0 (2)	0.1
Systemic manifestations, N	0	arthralgia 2	arthralgia 1, skin sclerosis 1 ^c	arthralgia 1	
Treatment, % (N)	0.0 (0)	0.0 (0)	0.0 (0)	0.0 (0)	N/A

p-values were obtained by analysis of variance for continuous value and chi-squared test for categorical value.

a: n=4

b: n=3

c: modified Rodnan total skin thickness score was 4, not classified as limited cutaneous systemic sclerosis

We agree that this finding is relevant in light of our observations regarding age-associated B cells (ABCs). Additional differential gene expression analysis using scRNA-seq data revealed that memory B cells from both the CENT+ SjD and SSA+CENT+SjD groups exhibited increased *ZEB2* expression, a transcription factor critical for ABCs differentiation, compared to the Sicca group (Fig. 3g). However, this increase was not observed in the SSA+ group. These findings contextualise our observations regarding the distinct B cell profiles in these patient subgroups.

Fig. 3g: Differentially expressed gene (DEG) analysis in memory and plasma cells (Plasma and *NR4A1*⁺*NFKB*⁺ plasma) across SjD subgroups compared to Sicca.

Volcano plots show \log_2 fold changes (\log_2FC) (x-axis) and $-\log_{10}$ adjusted p-values (y-axis) for DEGs between each SjD subgroup and Sicca controls in memory (left three panels) and plasma cells (right three panels). Upregulated genes (red) and downregulated genes (blue) are highlighted with selected key genes annotated. Genes not reaching significance are shown in grey.

We believe that the inclusion of Supplementary Table 1 and updates of the manuscript text significantly enhances the clarity and comprehensiveness of our manuscript, providing valuable context for our findings.

Results, Line 182, Page 6:

“A detailed summary of clinical and immunological characteristics is provided in Supplementary Table 1, showing distinct patterns across autoantibody subgroups. For example, CENT+ patients tended to be older and exhibited more frequent features such as Raynaud’s phenomenon and skin sclerosis, while SSA+ patients showed higher IgG levels. “

Results, Line 329, Page 9:

“To further investigate disease-specific transcriptional alterations in B cells, we performed differential gene expression analyses of memory and plasma B cell subsets across SjD subgroups compared to Sicca controls (**Figure 3g, Supplementary Table 2**). In memory B cells, the CENT+ and SSA+CENT+ groups showed upregulation of several genes involved in antigen presentation and survival, including *HLA-DRB1*, *CD40*, and *BCL2*. Notably, *ZEB2*, a transcription factor implicated in the differentiation toward ABCs³⁶, was significantly upregulated in both CENT+ and

SSA+CENT+ groups, but not in SSA+ alone. These findings highlight a distinct ABC-like transcriptional signature enriched in CENT+ and double-positive patients“

Discussion, Line 652, Page 19:

“Consistent with prior clinical observations, CENT-positive patients in our cohort tended to be older than CENT-negative patients (SSA+ or Sicca). This trend is particularly relevant given our transcriptional findings: CENT+ and SSA+CENT+ patients exhibited elevated expression of *ZEB2*, a transcription factor critical for the differentiation of ABCs, specifically within memory B cells, suggesting a distinct immunological profile associated with CENT positivity and older age. These data highlight the interplay between age, serological status, and B cell phenotypes in SjD pathogenesis, and support the hypothesis that CENT+ SjD represents a biologically distinct subset.”

Reviewer 1 Comment #4

- Figures and Legends:
 - While the figures and legends are well-prepared, the visualization of statistical results needs improvement for better interpretability. For example, in Figure 2I, the red square in the graph overlapping with the red circles is unclear. Use a different colour or symbol to distinguish regions with FDR < 0.05 more effectively.

We appreciate the reviewer’s helpful suggestion. In the revised version, we now highlight only the neighborhoods with FDR < 0.05 by coloring them, while leaving non-significant neighborhoods uncolored. We have also applied similar improvements to other figures throughout the manuscript to enhance overall readability and interpretability of the statistical visualizations. We hope these changes enhance clarity and ensure that significant regions are more clearly distinguished.

Revised Fig. 2I (Fig. 2h in the revised manuscript):

Minor Comments:

Reviewer 1 Comment #5

- Terminology:
 - Use "Sjögren’s disease" instead of "Sjögren’s syndrome" in line with the 2021 recommendation by Baer & Hammitt (1). This terminology has been endorsed by American and European patient groups and is now widely adopted in the literature. (Reference: Baer, A. N., & Hammitt, K. M. (2021). Sjögren’s disease, not syndrome. *Arthritis Rheumatol.*, 73, 1347–1348.)

We appreciate and agree with the reviewer's comment in terms of this terminological update. We have revised the manuscript accordingly by replacing all instances of "Sjögren's syndrome (SjS)" with "Sjögren's disease (SjD)" throughout the text.

Reviewer 1 Comment #6

- Autoantibody Overlap:
 - Include a visualization or table showing the overlap of autoantibodies among the three groups (anti-SSA+, anti-centromere+, and anti-SSA+/anti-centromere+) and other autoantibodies such as ANA, RF, and anti-La.

We thank the reviewer for this suggestion. As described in our response to Reviewer 1 Comment #3, we have updated Supplementary Table 1 to include detailed information regarding the overlap of autoantibodies (including ANA, RF, and anti-La) among the three groups (anti-SSA+, anti-centromere+, and anti-SSA+/anti-centromere+). We believe this updated table effectively addresses the reviewer's recommendation by clearly presenting the autoantibody profiles across patient subgroups.

Reviewer 1 Comment #7

- Ro52 and Ro60 Reactivity:
 - Specify whether patients in the anti-SSA+ and anti-SSA+/anti-centromere+ groups show seropositivity for Ro60, Ro52, or both. This detail would strengthen the interpretation of B cell reactivity and its relationship with autoantibody profiles.

As detailed in the revised Supplementary Table 1, we have now specified the seropositivity for Ro52, Ro60, or both, for patients within the anti-SSA+ and anti-SSA+/anti-CENT+ groups. We appreciate the reviewer's suggestion and believe this clarification enhances the depth of our discussion on the immunological features of these Sjögren's Disease subgroups.

Reviewer 2 (Remarks to the Author):

Inamo and colleagues in this manuscript aim at understanding the mechanisms underlying the association of certain antibodies with Sjögren's Syndromes (SS). SS diagnosis is based on essentially two types of antibodies, anti-SSA and anti-centromere. Certain aspects of SS overlap with the Sicca syndrome and systemic scleroderma. The field is certainly in need of a better understanding of the pathogenesis beyond descriptive clinical diagnoses and still very superficial antibody tests. As a matter of fact, misdiagnosis and underdiagnosis are quite common.

As a consequence, this paper is timely in its approach and quite interesting for the amount of data provided. The authors studied samples from salivary glands obtained from three different categories of patients with SS stratified on the basis of single or double positivity to SSA and anti-centromere antibodies and a group of control constituting of Sicca syndrome patients. A combination of multi-modal single cell technologies, including single cell RNA sequencing, T cell and B receptor sequencing and special transcriptomics was employed. The authors tried to integrate the various datasets in a cogent picture. The approach is certainly laudable and leads to a very interesting data-heavy, perhaps too heavy, manuscript, which describes a litany of findings, separated by cell type. While some of these findings were previously described, the richness and scope of the data lend to the appreciation of this work as a potentially very useful reference paper.

The limitations of this manuscript, some of them honestly acknowledged by the authors, are often inherent to the technologies used. For instance, TCR and BCR sequencing lend to data which can be interpreted only through speculation, as demonstrated also by the authors, by identification of clones, which are not necessarily specific for the antigens which have stratified the patient populations.

Other limits relate to the spatial transcriptomic methodology which was employed (Visium), which does not have a single cell resolution and the results are therefore fuzzy. As a consequence, the manuscript message is less incisive than what it could be, with most of the interpretation of the data speculative rather than fact-driven. As also underscored by the authors, such limitations are also increased by the relatively small sample size.

However, again, there is a certain value in this paper as a reference.

We sincerely appreciate the positive feedback and the thoughtful evaluation of our work. We are grateful for the reviewer's recognition of our efforts to integrate multi-modal single-cell and spatial transcriptomic analyses, and for highlighting the potential value of our manuscript as a useful reference.

We fully agree that determining the antigen specificity of T and B cell clones based solely on sequence data remains a major technical limitation of current high-throughput immune repertoire analysis. While clonal expansion suggests potential antigen-driven selection, definitive specificity—particularly toward disease-defining autoantigens such as SSA/Ro or centromere—is difficult to confirm without functional validation.

As the reviewer points out, this limitation is indeed acknowledged in the manuscript. To provide further context, in our prior study (*Ann Rheum Dis.* 2020;79(1):150-158. PMID: 31611218), we performed single-cell cloning of antibody-secreting cells from anti-Ro60-positive patients and found that 15.6% of these antibodies recognized Ro60. Although this frequency is markedly higher than that observed in peripheral blood (<1%), the majority of antibodies still lacked reactivity to canonical autoantigens. Similarly, the frequency of centromere-specific antibodies (abbreviated as 'ACA' in the following table) in a single CENT+ patient in that cohort was 38.5%. These findings support the notion that autoantigen-specific clones are enriched at the site of inflammation, yet constitute only a subset of the total expanded repertoire.

Supplementary Tables S4 and S5 in our previously published paper (*Annals of the Rheumatic Diseases.* 2020; 79(1): 150–158) (PMID: 31611218)

[editorial note: tables redacted]

In the current study, we attempted to assess the antigen reactivity of the most expanded BCR clones using intracellular staining against 293T cells. While a few clones were confirmed to be reactive, others were negative, which may reflect either a true lack of autoreactivity or limitations in detection sensitivity. As the reviewer rightly notes, clone expansion alone does not guarantee specificity for the stratifying autoantigen. Indeed, bystander activation, non-classical autoantigens, or responses to microbial components may contribute to clonal selection in the inflammatory microenvironment.

We have now clarified this point more explicitly in the revised manuscript.

Discussion, Line 661, Page 19:

“In our previous study¹¹, we found that 15.6% and 38.5% of antibody-secreting cells infiltrating the salivary glands of patients positive for serum anti-Ro60 or CENT, respectively, produced Ro60- or CENT-specific antibodies. In the present study, we further investigated the antigen specificity of B cells at the lesion site by generating recombinant antibodies from the most clonally expanded sequences within each of the top five BCR trees. Although the experimental methods differ and

a direct comparison is therefore not possible, we confirmed the presence of autoreactive antibodies at a frequency consistent with previous reports.”

We sincerely thank the reviewer for thoughtful critique, which helped us to clarify the limitations and interpretation of our immune repertoire analysis.

We will address the reviewer’s specific points in detail below.

Reviewer 2 Comment #1

- What could be done to improve the paper with available data relates to the improvement of correlative analyses, by providing, for instance, network analyses. The interaction between stroma and immune cell is a very important finding, but its reporting remains somewhat shallow, both in the depth of the data produced and for their interpretation. Specifically, it would be much more informative to characterize functionally the interactions between fibroblasts and infiltrating immune cells, showing in much more detail what they express and providing more detailed pictures, possibly of direct mechanistic relevance. Indeed, the divide across auto-antibody profiles represents the most original component of this paper, and could be exploited a lot better, with more in-depth analysis of available data, and perhaps some limited additional mechanistic and validation studies.

We appreciate the reviewer’s suggestion for deeper correlative and network analyses. In response, we performed an additional cell–cell communication analysis using our scRNA-seq dataset and the CellChat pipeline. This enabled us to map ligand–receptor pairs among the identified cell clusters, revealing that *THY1*⁺ fibroblasts are critical mediators of pro-inflammatory signals in the salivary gland microenvironment. Specifically, the *C3-ITGB2* interaction previously detected by our spatial transcriptomics data was confirmed here to be mediated between classical/non-classical monocytes and dendritic cells as receiver (*ITGB2*) and *THY1*⁺ fibroblasts as sender (*C3*), underlining the importance of fibroblast–immune cell cross-talk.

In this analysis, we also observed disease-specific alterations in T cell–fibroblast communication. In CENT+ SjD, a broader range of cytokine and adhesion molecule interactions was detected, including *TNF–TNFRSF1A*, a pro-inflammatory signaling axis that promotes NF-κB–mediated gene expression; *SIRPG–CD47*, an immune-regulatory interaction implicated IFN-γ secretion by chronically activated T cells; and *CD46–JAG1*, a ligand–receptor pair that represents a crosstalk between the complement system and Notch signaling, promoting Th1 responses. These interactions were absent or less pronounced in SSA+ SjD. These findings suggest that global T cell activation in CENT+ patients may contribute to local fibroblast activation, a hallmark of tissue remodeling and fibrosis, further supporting the notion of subgroup-specific pathogenic mechanisms.

In the revised manuscript, we have added this new figure and updated the manuscript text accordingly. We hope these enhancements address the reviewer’s concerns by offering a functionally informed view of the fibroblast–immune cell interactions in Sjögren’s disease.

a

b

Extended Data Fig. 10: Cell-cell communication analysis highlights the central role of *THY1*⁺ fibroblasts in SjD.

a-b, Heatmaps depicting predicted ligand-receptor interactions in which *THY1*⁺ fibroblasts are either the signal-sending (a) or signal-receiving (b) population within each subgroup. Rows list

the individual ligand–receptor pairs, while columns represent the interacting cell clusters. Color intensity indicates the communication probability. The size/shape of the dots denotes statistical significance.

Results, Line 561, Page 16:

“In order to further delineate the *THY1*⁺ fibroblasts–immune cell interactions, suggested by our spatial transcriptomics findings, we performed a cell–cell communication analysis on our scRNA-seq data. This revealed that *THY1*⁺ fibroblasts exhibited a number of the ligand–receptor pairs implicated *THY1*⁺ fibroblasts either as the signal source or target, underscoring their pivotal role in coordinating immune responses (**Extended Data Figure 10**). Among these, the *C3-ITGB2* axis emerged again as an interaction signal between monocytes and *THY1*⁺ fibroblasts.

In this analysis, we also observed disease-specific alterations in T cell–fibroblast communication. In CENT+ SjD, a broader range of cytokine and adhesion molecule interactions was detected, including *TNF–TNFRSF1A*, a pro-inflammatory signaling axis that promotes NF-κB-mediated gene expression⁴⁴; *SIRPG–CD47*, an immune-regulatory interaction implicated IFN-γ secretion by chronically activated T cells⁴⁵; and *CD46–JAG1*, a ligand–receptor pair that represents a crosstalk between the complement system and Notch signaling, promoting Th1 responses⁴⁶. These interactions were absent or less pronounced in SSA+ SjD. These findings suggest that global T cell activation in CENT+ patients may contribute to local fibroblast activation, a hallmark of tissue remodeling and fibrosis, further supporting the notion of subgroup-specific pathogenic mechanisms.

Collectively, these findings highlight the complexity of cell-cell communication within the SjD salivary gland microenvironment and solidify the central role of *THY1*⁺ fibroblasts as potential drivers of inflammation and tissue pathology.”

- A final note relates to the figures, which are way too busy, landing, or better, deducting from clarity, particularly whenever images are provided.

We appreciate the reviewer’s helpful suggestion. In the revised manuscript, we modified figures to improve visualization and clarity. For instance, Figure 2I (and similar figures), we now highlight only the neighborhoods with *FDR* < 0.05 by coloring them, while leaving non-significant neighborhoods uncolored. We have also applied similar improvements to other figures throughout the manuscript to enhance overall readability and interpretability of the statistical visualizations. We hope these changes enhance clarity and ensure that significant regions are more clearly distinguished.

Revised Fig. 2I (Fig. 2h in the revised manuscript):

h

Reviewers' comments:

Reviewer 3 (Remarks to the Author):

The paper by Inamo J et al describes an interesting aim to understand the molecular and cellular derangements of autoantibody defined Sjogren's disease. The authors should be complimented for their approach using supplementary single cell transcriptomic techniques, clonality analysis and assessment of auto-antibody producing clones. The authors indicate that "redefinition of SjD based on autoantibody status and underlying molecular mechanisms could pave the way for development of targeted therapies". Although this could significantly help it is questionable whether the data that are presented are paving such a way. Overall, robust (statistically significant) and reliable findings between CENT+ and SSA+ patients and direct associations between the autoimmunity features and clinical features are lacking. There are several major and minor concerns:

We greatly appreciate the reviewer's thorough review of our manuscript and kind words regarding our approach. We acknowledge the reviewer's concerns regarding the statistical robustness and the direct associations between autoimmunity features and clinical outcomes.

To address these valid points and strengthen the statistical power of our findings, **we have increased the number of CENT+ SjD samples for scRNA-seq analysis from the initial 4 to 7**. This expansion of our dataset is aimed at enhancing the reliability of our findings and enabling more robust statistical comparisons between the different SjD subgroups.

We will address the reviewer's points in detail below.

Major Comments:

Reviewer 3 Comment #1

- Direct associations of autoimmunity and clinical features are missing. Differences in inflammatory markers may well be associated with heterogeneity of the patients selected. In this respect in the Supplementary table I the retrospectively collected clinical features are shown, showing that in SSA+ patients overall have more severe inflammation and overall are characterized by a higher generalized B cell hyperactivity. In addition, the clinical features of the patients are lacking in terms of systemic disease activity/organ involvement as captured by ESSDAI scores. Because CENT+ patients may also represent SSc patients, a different clinical entity, it is important to have in depth clinical characterization including skin scores (eg MRSS), etc. Unfortunately, this is lacking,

making the direct relationship of the observed cellular and molecular features and autoimmunity unclear.

We thank the reviewer for this thoughtful and important comment. We agree that in-depth clinical characterization is critical for interpreting molecular and cellular heterogeneity in SjD subgroups.

In the revised Supplementary Table 1, we have included clinical and serologic features across patient groups, including markers of systemic inflammation (e.g., serum IgG, complement levels), autoantibody profiles (anti-Ro60, anti-Ro52, anti-La, ANA, RF), and organ involvement (e.g., interstitial lung disease, Raynaud’s phenomenon, cutaneous sclerosis with MRSS, arthralgia). These clinical observations support the notion that CENT+ SjD patients exhibit a distinct phenotype with more vascular and fibrotic manifestations, in contrast to the SSA+ group which displayed higher IgG levels, consistent with increased B cell hyperactivity. Furthermore, ESSDAI scores were provided for all SjD subgroups, although scores were generally low across all groups, likely reflecting the mild systemic activity of the disease in our cohort.

We believe the breadth of serological, histopathological, and systemic involvement indicators included allows for a meaningful interpretation of the relationship between autoantibody-defined subgroups and the observed molecular and cellular profiles.

Results, Line 182, Page 6:

“A detailed summary of clinical and immunological characteristics is provided in Supplementary Table 1, showing distinct patterns across autoantibody subgroups. For example, CENT+ patients tended to be older and exhibited more frequent features such as Raynaud’s phenomenon and skin sclerosis, while SSA+ patients showed higher IgG levels. “

Supplementary Table 1a. Characteristics of subjects for paired scRNA-seq and repertoire data.

	Sicca	SSA+ SjD	CENT+ SjD	SSA+CENT+ SjD	p-values
N	8	8	7	4	
Age, [IQR]	54.5 [44.8, 66.8]	50.0 [42.5, 77.8]	70.0 [63.0, 78.0]	62.0 [52.0, 71.8]	0.44
Female, % (N)	75.0 (6)	100.0 (8)	85.7 (1)	75.0 (3)	0.50
Duration, month [IQR]	9.5 [2.0, 15.0]	16.0 [3.8, 31.5]	12.0 [7.0, 48.0]	18.0 [11.0, 24.0]	0.62
IgG, mg/dl [IQR]	1567.0 [1320.2, 1747.5]	1333.5 [961.2, 1521.8]	1187.0 [1048.0, 1259.0]	1744.0 [1484.5, 2041.5]	0.096
Ant-Ro60 antibody positive, % (N)	0.0 (0) ^a	100.0 (8)	0.0 (0)	100.0 (4)	<0.0001
Anti-Ro52 antibody positive, % (N)	0.0 (0) ^b	37.5 (3)	14.3 (1)	25.0 (1)	0.36
Anti-SSB antibody positive, % (N)	0.0 (0)	37.5 (3)	14.3 (1)	50.0 (2)	0.058

ANA positive, % (N)	25.0 (2)	50.0 (4)	100.0 (7)	100.0 (4)	0.0074
Rheumatoid factor positive, % (N)	37.5 (3)	37.5 (3)	14.3 (1)	50.0 (2)	0.62
C3, mg/dL [IQR]	95.5 [81.0, 108.0]	82.0 [69.0, 97.0] ^a	105.0 [96.0, 112.5] ^b	101.5 [89.8, 117.0]	0.052
C4, mg/dL [IQR]	21.5 [12.3, 25.0]	25.0 [18.0, 28.0] ^a	22.5 [20.0, 26.0] ^b	28.5 [23.3, 30.0]	0.29
ESSDAI score, index [IQR]	NA	0.5 [0.0, 6.25]	0.0 [0.0, 2.0]	0.5 [0.0, 3.3]	0.71
Focus score [IQR]	0.0 [0.0, 0.0]	0.4 [0.0, 2.5]	0.0 [0.0, 4.2]	4.4 [0.5, 6.8]	0.040
Rainaud's phenomenon positive, % [N]	0.0 (0)	12.5 (1)	42.9 (3)	50.0 (2)	0.16
Systemic manifestations, N	0	ILD 1, CV 1, arthralgia 1	ILD 1, skin sclerosis 2 ^d	arthralgia 1	
Treatment, % (N)	0.0 (0)	12.5 (1) ^e	0.0 (0)	0.0 (0)	0.48

p-values were obtained by analysis of variance for continuous value and chi-squared test for categorical value.

a: n=7

b: n=6

c: n=3

d: Two patients showed skin sclerosis limited to the fingers (mRSS = 4). One of them met the classification criteria for limited cutaneous systemic sclerosis based on the combination of Raynaud's phenomenon and anti-centromere antibody positivity, while the other did not meet the criteria due to the absence of Raynaud's.

e: prednisolone 50 mg/day, hemodialysis, intravenous cyclophosphamide 1000 mg/month, rituximab 375 mg/m² twice 3 months prior to lip biopsy, hemodialysis for thrombotic microangiopathy and cryoglobulinemic vasculitis

ILD; interstitial lung disease, CV; cryoglobulinemic vasculitis

Supplementary Table 1b. Characteristics of subjects for Visium data.

	Sicca	SSA+ SjD	CENT+ SjD	SSA+CENT+ SjD	p-values
N	5	6	5	4	
Age, [IQR]	41.0 [39.0, 48.0]	47.5 [41.8, 63.0]	59.0 [54.0, 67.0]	62.0 [52.0, 71.8]	0.15
Female, % (N)	100 (5)	83.3 (5)	100 (5)	75.0 (3)	0.49

Duration, month [IQR]	24.0 [2.0, 36.0]	36.0 [9.0, 60.0]	60.0 [24.0, 168.0]	18.0 [11.0, 24.0]	0.62
IgG, mg/dl [IQR]	1000.5 [935.8, 1177.5]	1965.5 [1509.2, 2319.0]	1226.0 [1171.0, 1253.0]	1744.0 [1484.5, 2041.5]	0.04
Ant-Ro60 antibody positive, % (N)	0.0 (0)	100.0 (6)	0.0 (0)	100.0 (4)	0.0002
Anti-Ro52 antibody positive, % (N)	0.0 (0) ^a	75.0 (3) ^a	0.0 (0) ^a	25.0 (1)	0.42
Anti-SSB antibody positive, % (N)	0.0 (0)	66.7 (4)	0.0 (0)	50.0 (2)	0.031
ANA positive, % (N)	20.0 (1)	100.0 (6)	100.0 (5)	100.0 (4)	0.0018
Rheumatoid factor positive, % (N)	0.0 (0) ^a	83.3 (5)	25.0 (1) n=4	50.0 (2)	0.057
C3, mg/dL [IQR]	94.0 [76.8, 109.0] ^a	105.5 [78.0, 143.5] ^a	84.0 [77.0, 87.0] ^b	101.5 [89.9, 117.0]	0.57
C4, mg/dL [IQR]	19.0 [16.3, 21.8] ^a	18.0 [15.8, 22.5] ^a	14.0 [14.0, 25.0] ^b	28.5 [23.3, 30.0]	0.077
ESSDAI score, index [IQR]	NA	1.0 [0.0, 4.3]	0.0 [0.0, 1.0]	0.5 [0.0, 3.3]	0.48
Focus score [IQR]	0.0 [0.0, 0.0]	2.1 [1.2, 5.7]	4.0 [2.4, 4.7]	4.4 [0.5, 6.8]	0.021
Rainaud's phenomenon positive, % [N]	0.0 (0)	0.0 (0)	40.0 (2)	50.0 (2)	0.1
Systemic manifestations, N	0	arthralgia 2	arthralgia 1, skin sclerosis 1 ^c	arthralgia 1	
Treatment, % (N)	0.0 (0)	0.0 (0)	0.0 (0)	0.0 (0)	N/A

p-values were obtained by analysis of variance for continuous value and chi-squared test for categorical value.

a: n=4

b: n=3

c: modified Rodnan total skin thickness score was 4, not classified as limited cutaneous systemic sclerosis

Reviewer 3 Comment #2

- Although there is an enrichment of certain pathways in T and B cell types and tissue cells in the SSA+ and CENT+ groups this study does not show significant differences between these 2 groups in a direct comparison. Figures 2-7. Single cell data are studied at the group level using pooled analyses of single cells. It is unclear whether the molecular and cellular features of the individual donors are associated with the clinical features.

We thank the reviewer for this insightful comment. We fully acknowledge the importance of directly comparing molecular and cellular profiles between SSA+ and CENT+ SjD groups, and of considering inter-individual variability and its relationship to clinical phenotypes.

To address this, we have taken several steps to improve both the robustness and interpretability of our analysis:

1. **Increased CENT+ sample size:** We expanded the number of CENT+ SjD samples included in the scRNA-seq analysis from 4 to 7, thereby enhancing statistical power and enabling more reliable subgroup comparisons.
2. **Direct comparison of SSA+ vs. CENT+ groups:** With the expanded dataset, we re-performed differential abundance analyses across immune cell populations between the SSA+ and CENT+ groups using miloR, a well-established framework that models fixed effects (e.g., group differences) while accounting for random effects (e.g., donor variability) through generalized linear mixed models. This approach is particularly suited for single-cell data, where batch effects and subject-level variability can confound true biological signals.

These analyses revealed that:

- Tph/Tfh cells were expanded in both SSA+ and CENT+ SjD compared to Sicca, with CENT+ SjD showing a trend toward greater expansion, suggesting a stronger local T cell activation.

Fig. 2h: Differential abundance profiles of T cell populations in SjD subtypes.

Beeswarm plots showing the distribution of log₂-fold change (logFC) in neighborhoods in different cell type clusters in T cells. Each plot compares the abundance in SjD overall, SSA+, CENT+, and SSA+CENT+ subtypes versus Sicca or each other. Significant changes (false discovery rate (FDR) < 0.05) are highlighted, indicating enriched or depleted in each case.

- Memory B cells were significantly increased in CENT+ SjD (and SSA+CENT+ SjD) compared to Sicca, whereas no such increase was observed in the SSA+ group.

Fig. 3h: Differential abundance profiles of B/plasma cell populations in SjD subtypes.

Beeswarm plots showing the distribution of log₂-fold change (logFC) in neighborhoods in different cell type clusters

in T cells. Each plot compares the abundance in SjD overall, SSA+, CENT+, and SSA+CENT+ subtypes versus Sicca or each other. Significant changes (FDR < 0.05) are highlighted, indicating enriched or depleted in each case.

These results suggest that CENT+ SjD may exhibit more pronounced local B cell and T cell activation within the salivary glands, offering insights into subgroup-specific immunopathology.

We have added explanatory text describing these findings in the Results section in the revised manuscript.

Results, Line 354, Page 10:

“In comparing analysis of cell type abundance, we observed memory B cells were increased in CENT+ subtype compared to Sicca, while plasma cells were more abundant in SSA+ group (Figure 3h).”

In terms of association between cellular features and clinical features, we performed a correlation analysis between cell type frequencies and clinical variables, including age, IgG titer, disease duration, and focus score. These results are now presented in Extended Data Fig. 6.

Specifically, we calculated Spearman’s correlation coefficients between the relative abundance of both broad cell types (panel a) and fine-scale cell clusters (panel b) and clinical features across all patients and within each autoantibody-defined subgroup (SSA+ SjD, CENT+ SjD, SSA+CENT+ SjD).

Key findings include:

- Across all SjD patients, the frequency of Tph/Tfh and Treg cells positively correlated with focus score, suggesting their role in driving lymphocytic infiltration.
- A consistent trend observed across groups was that increased age correlated with reduced acinar cell abundance and increased ductal cells, potentially reflecting age-associated glandular atrophy and functional decline (e.g., reduced salivary secretion).
- IgG titer showed a positive correlation with *GZMK*⁺ CD8⁺ T cells in both SSA+ and CENT+ SjD subgroups, but not with B or plasma cell frequencies. This suggests that systemic IgG elevation may not directly reflect local B/plasma cell abundance in labial salivary glands. This discrepancy may be due to sampling limitations of minor glands or reflect the contribution of activated B cells in other anatomical sites (e.g., lymph nodes).
- In the CENT+ subgroup, unique positive correlations were observed between *THY1*⁺ fibroblast/endothelial cells and age, suggesting that aging may contribute to the expansion or activation of stromal populations involved in tissue remodeling and fibrosis.

Together, these revisions allow for a clearer interpretation of how specific cellular programs may relate to clinical autoantibody-defined phenotypes. We have updated figures and the manuscript text accordingly.

We thank the reviewer again for prompting this important clarification and additional analysis, which we believe have substantially strengthened the manuscript.

Extended Data Fig. 6: Correlation between cell-type frequencies and clinical features.

Heatmaps showing Spearman's correlations between fine-grained immune and stromal cell clusters and clinical variables. T, B/plasma, and other leukocyte subsets are normalized within their respective parent lineages, while tissue cells are normalized within total tissue-derived cells. Statistically significant correlations are marked: * $P < 0.05$ (nominal), ** $P < 0.05$ (Benjamini–Hochberg adjusted). Color scale represents the strength and direction of the correlation (red: positive, blue: negative).

Results, Line 421, Page 12:

“Correlation of clinical variables with immune and tissue cell composition across SjD subtypes

To evaluate whether cellular features are associated with clinical parameters, we performed correlation analyses between the frequency of cell clusters and clinical variables including age, disease duration, serum IgG titer, and histological focus score (**Extended Data Figure 6**). We observed consistent patterns across all patients, including a negative correlation between acinar (serous and mucous) cell frequency and age, and a corresponding positive correlation between ductal epithelial cell frequency and age. These trends may reflect age-related loss of secretory function and glandular remodeling.

Among immune subsets, the proportion of Tph/Tfh and Treg cells within the T cell compartment positively correlated with focus score in the overall SjD patients, suggesting their contribution to lymphocytic infiltration in inflamed glands. Interestingly, the IgG titer was not significantly correlated with the frequency of B or plasma cells in any of the subgroups. This suggests that systemic B cell activation may reflect activity occurring outside the salivary gland, such as lymphoid tissues.

Subtype-specific associations were also noted. In the CENT+ subgroup, *THY1*⁺ fibroblast and endothelial cell frequencies showed a unique positive correlation with age, suggesting that aging in CENT+ SjD patients may contribute to the expansion or activation of stromal populations involved in tissue remodeling and fibrosis.

These findings collectively highlight both shared and subgroup-specific relationships between cell composition and clinical phenotypes in SjD.”

Reviewer 3 Comment #3

- From the up-regulated T cell pathways that are shown in Fig 2G it appears that CENT+ patients have the lowest nr of pathways (unique pathways 66, 8, 308 and 101 for SICCA, CENT+, SSA+ and CENT+SSA+ patients, respectively). This indicates that overall T cells of CENT+ patients are characterized by reduced T cell activation. This is not reflected in the analysis of the T cell subsets in pathway upregulation (Figure 2G and 2H) and is a puzzling discrepancy. In this respect, one would think that eg. Tph/Tfh cells, highly activated cells shown to produce a considerable number of cytokines, to show some enrichment of these pathways, which however is not the case (Fig. 2H). The authors should also (next to T cell subsets) look at whether the net/global T cell activation is related to immunopathological features (eg activation of tissue cells, APCs, B cell activity, fibrosis, destruction). This could pave the way to better understanding the immune-driven differences between the groups.

We thank the reviewer for this thoughtful and constructive comment. We agree that in our original pathway-level analysis (the initial version of Figure 2G–H), the limited number of significantly enriched T cell pathways in CENT+ SjD could give the impression of globally reduced T cell activation. However, as the reviewer rightly noted, this appears discordant with the presence of activated T cell subsets such as Tph/Tfh cells, which are known to express cytokines and costimulatory molecules.

To address this, we performed a more granular gene-level differential expression analysis using the scRNA-seq data. Specifically, we compared each disease group (SSA+, CENT+, SSA+CENT+) against the Sicca group, separately for CD4⁺ and CD8⁺ T cells (Fig. 2g in the revised manuscript). This approach avoids dilution of key signals that may occur in pathway-level analyses, and allows us to directly identify gene-level differences specific to each subgroup.

Importantly, this analysis revealed that in CENT+ SjD, there is increased expression of Tph/Tfh-related genes, such as *MAF* and *ICOS*, in CD4⁺ T cells consistent with the reviewer’s expectations and supporting the presence of an activated T helper phenotype. Additionally, the expression of other activation markers (e.g., *CXCR4*, *CD69*, *TIGIT*) was elevated across both SSA+ and CENT+ groups. To provide a more comprehensive overview, we now include the full list of significant genes and pathways (adjusted *p*-value < 0.05) as Supplementary Table 2-3 for CD4⁺ and CD8⁺ T cells, respectively.

Fig. 2g: Differentially expressed gene (DEG) analysis in CD4⁺ and CD8⁺ T cells across Sjd subgroups compared to Sicca.

Volcano plots show \log_2 fold changes (\log_2 FC) (x-axis) and $-\log_{10}$ adjusted p-values (y-axis) for DEGs between each Sjd subgroup and Sicca controls in CD4⁺ (left three panels) and CD8⁺ T cells (right three panels). Upregulated genes (red) and downregulated genes (blue) are highlighted with selected key genes annotated. Genes not reaching significance are shown in grey.

As for the relationship between T cell activation and immunopathological features, we examined cell–cell communication networks using CellChat, focusing on interactions between T cell subsets and *THY1*⁺ fibroblasts, which represent activated tissue stromal cells based on our spatial transcriptomic analysis. In this analysis, we also observed disease-specific alterations in T cell–fibroblast communication. In CENT+ Sjd, a broader range of cytokine and adhesion molecule interactions was detected, including *TNF–TNFRSF1A*, a pro-inflammatory signaling axis that promotes NF- κ B-mediated gene expression; *SIRPG–CD47*, an immune-regulatory interaction implicated IFN- γ secretion by chronically activated T cells; and *CD46–JAG1*, a ligand–receptor pair that represents a crosstalk between the complement system and Notch signaling, promoting Th1 responses. These interactions were absent or less pronounced in SSA+ Sjd. These findings suggest that global T cell activation in CENT+ patients may contribute to local fibroblast activation, a hallmark of tissue remodeling and fibrosis, further supporting the notion of subgroup-specific pathogenic mechanisms.

Together, these results offer a clearer picture of the transcriptional activation programs in CENT+ and SSA+ patients and resolve the apparent discrepancy between pathway-level summaries and cell-type-specific gene activation. We thank the reviewer for prompting this deeper analysis.

a

b

Extended Data Fig. 10: Cell-cell communication analysis highlights the central role of *THY1*⁺ fibroblasts in SjD.

a-b, Heatmaps depicting predicted ligand–receptor interactions in which *THY1*⁺ fibroblasts are either the signal-sending (**a**) or signal-receiving (**b**) population within each subgroup. Rows list the individual ligand–receptor pairs, while columns represent the interacting cell clusters. Color intensity indicates the communication probability. The size/shape of the dots denotes statistical significance.

Results, Line 561, Page 16:

“In order to further delineate the *THY1*⁺ fibroblasts–immune cell interactions, suggested by our spatial transcriptomics findings, we performed a cell–cell communication analysis on our scRNA-seq data. This revealed that *THY1*⁺ fibroblasts exhibited a number of the ligand–receptor pairs implicated *THY1*⁺ fibroblasts either as the signal source or target, underscoring their pivotal role in coordinating immune responses (**Extended Data Figure 10**). Among these, the *C3-ITGB2* axis emerged again as an interaction signal between monocytes and *THY1*⁺ fibroblasts.

In this analysis, we also observed disease-specific alterations in T cell–fibroblast communication. In CENT+ SjD, a broader range of cytokine and adhesion molecule interactions was detected, including *TNF–TNFRSF1A*, a pro-inflammatory signaling axis that promotes NF-κB-mediated gene expression⁴⁴; *SIRPG–CD47*, an immune-regulatory interaction implicated IFN-γ secretion by chronically activated T cells⁴⁵; and *CD46–JAG1*, a ligand–receptor pair that represents a crosstalk between the complement system and Notch signaling, promoting Th1 responses⁴⁶. These interactions were absent or less pronounced in SSA+ SjD. These findings suggest that global T cell activation in CENT+ patients may contribute to local fibroblast activation, a hallmark of tissue remodeling and fibrosis, further supporting the notion of subgroup-specific pathogenic mechanisms.

Collectively, these findings highlight the complexity of cell-cell communication within the SjD salivary gland microenvironment and solidify the central role of *THY1*⁺ fibroblasts as potential drivers of inflammation and tissue pathology.“

Reviewer 3 Comment #4

- Related to the previous concern, it is helpful to show the VENN diagrams for B cells, tissue cells, APCs (figure 3 and 4). This would help to appreciate the total nr of pathways that are associated with T cell dysregulation.

We appreciate the reviewer’s suggestion to include Venn diagrams to help visualize the overlap and distribution of pathway enrichment across cell types. To address this point and improve interpretability, in the revised version, we performed subset-specific differentially expressed gene analysis and pathway enrichment analysis within the major cell types across SjD subgroups compared to Sicca controls. These results are now provided as Supplementary Tables 2-3, which include the full list of significantly enriched genes and pathways (adjusted *p*-value < 0.05) for each subset.

We believe this approach allows for a clearer understanding of how distinct immune and stromal subsets contribute to immunopathology in each subgroup, and we thank the reviewer for encouraging us to enhance the clarity and depth of our pathway-level analysis.

Reviewer 3 Comment #5

- Abundance of non-Ro-recognizing clones in SSA+ patients: It is surprising that in SSA+ patients, the most abundant clones do not seem to recognize Ro, which one would expect

to define this population. This is particularly striking for patients 5, 7, and 8, where these non-Ro-recognizing clones are highly abundant. I could not find a discussion addressing this point in the manuscript. Do the authors have any data or hypotheses on what these clones might recognize? In addition, why does only 1 out of 3 (33%) CENT+ patients and 3 out of 4 CENT+SSA+ patients (75%) show CENT+ B cell clones. In this respect the authors should study if this is not just a reflection of generalized B cell hyperactivity that is often associated with SSA+ patients. In fact studying the relationship between T cell derangements and generalized B cell activity could reveal important immunopathological mechanisms.

We thank the reviewer for this insightful and important comment. We have long been interested in the antigen specificity of B cells within the salivary glands, and in our previous study (*Ann Rheum Dis.* 2020;79(1):150-158., PMID: 31611218, Supplementary table S4,S5, as shown below), we analyzed antibody-secreting cells randomly selected from the salivary glands of patients with anti-Ro60 antibody-positive serum. In that cohort, the proportion of Ro60-specific antibodies was approximately 15.6% (23 Ro60-specific antibodies out of 147 total antibodies). For anti-centromere-antibody (abbreviated as ACA here), only one ACA-positive sample was included in that study, and the frequency of ACA-reactive antibodies was 38.5%.

Supplementary Tables S4 and S5 in our previously published paper (*Annals of the Rheumatic Diseases.* 2020; 79(1): 150–158) (PMID: 31611218)

[editorial note: tables redacted]

These results, along with the current study, suggest that while disease-specific autoantibodies are indeed enriched at the site of inflammation compared to peripheral blood (where such specificities are typically <1%), not all B cells within the lesion necessarily recognize canonical autoantigens. This may reflect the heterogeneity of antigen targets or the presence of non-classical autoantigens. Additionally, bystander activation or responses to non-self antigens (e.g., viral or commensal-derived) cannot be excluded.

In the present study, we performed intracellular staining against 293T cells for all antibodies that were assessed. Notably, the top BCR clones from patients 5, 7, and 8 were negative. Given the relatively low sensitivity of intracellular staining, we cannot conclusively determine the absence of autoreactivity, but at the very least, these antibodies do not recognize the Ro, La, or centromere antigens investigated in this study.

As for two patients of the CENT-positive group, it may seem unexpected that CENT-reactive antibodies were not detected; however, the top 5 BCR trees account for less than 5% of the total B cell population. Considering our previous data, the lack of CENT detection in this subset is not implausible.

Interestingly, in the SSA+CENT+ group, the top 5 antibodies appeared to preferentially target either CENT or SSA, suggesting that in double-positive patients, one specificity may become dominant. While this observation is scientifically intriguing, we acknowledge that the sample size is still small and therefore refrain from making definitive conclusions at this point.

Accordingly, we have added the following discussion to the manuscript:

Discussion, Line 661, Page 19:

“In our previous study¹¹, we found that 15.6% and 38.5% of antibody-secreting cells infiltrating the salivary glands of patients positive for serum anti-Ro60 or CENT, respectively, produced Ro60- or CENT-specific antibodies. In the present study, we further investigated the antigen specificity of B cells at the lesion site by generating recombinant antibodies from the most clonally expanded sequences within each of the top five BCR trees. Although the experimental methods differ and a direct comparison is therefore not possible, we confirmed the presence of autoreactive antibodies at a frequency consistent with previous reports.”

Reviewer 3 Comment #6

- Why did the authors not compare CENT+ and SSA+ patients in Figure 4 and switch from this comparison of patients to sicca vs all pSS patients in Figure 5? To link all the observed molecular and cellular changes it would be very valuable/necessary to have the same comparisons.

We thank the reviewer for this important observation. We agree that consistent comparisons across figures are essential to fully integrate molecular and spatial findings.

To address this, we have included CENT+ vs. SSA+ comparisons for individual tissue cell clusters, myeloid and NK cell subsets in Extended Data Figure 5.

Extended Data Fig. 5: Differential abundance profiles of tissue and other leukocyte cell populations in SjD subtypes.

Beeswarm plots showing the distribution of log₂-fold change (logFC) in neighborhoods in different cell type clusters in tissue cells (top), and myeloid and NK cells (bottom). Each plot compares the abundance in SjD overall, SSA+, CENT+, and SSA+CENT+ subtypes versus Sicca or each other. Significant changes (FDR < 0.05) are highlighted, indicating enriched or depleted in each case.

Results, Line 417, Page 12:

“In tissue cells, mucous acini were more abundant in SSA+ SjD, whereas serous acini were more abundant in CENT+ SjD when comparing SSA+ vs. CENT+ SjD, warranting validation by immunostaining experiments using more samples (**Extended Data Figure 5**).”

We have also performed a direct comparison between CENT+ and SSA+ patients using the Visium spatial transcriptomics data. This analysis revealed a modest but statistically significant increase in spatial cluster 6 in CENT+ SjD compared to SSA+ SjD. As shown in Figure 5e, this cluster is composed predominantly of tissue-resident stromal cells, but also includes a minor proportion of T cells and myeloid cells. The expansion of this cluster in CENT+ patients may reflect greater local activation of immune cells, which is consistent with our scRNA-seq findings showing comparable or even higher expansion of Tph/Tfh cells in CENT+ SjD relative to SSA+.

We have included these new results in Extended Data Figure 7, and we have updated the manuscript text accordingly. We thank the reviewer for prompting this additional analysis, which helps further align our spatial and single-cell findings across disease subgroups.

Extended Data Fig. 7: Differential abundance profiles across spatial clusters in SjD subtypes.

Neighborhood graph of spatial regions and Beeswarm plots showing the distribution of log₂-fold change (logFC) in neighborhoods in different spatial clusters in salivary glands. Colors indicate the logFC between case and controls. Neighborhoods that increased in case are shown in red.

Results, Line 465, Page 13:

“In the direct comparison between CENT+ and SSA+ patients, spatial cluster 6 was modestly but significantly expanded in CENT+ SjD (**Extended Data Figure 7**). This cluster was composed primarily of tissue-resident stromal cells, with a minor contribution from T cells and myeloid cells.

The expansion of this cluster in CENT+ patients may reflect enhanced local immune activation within the tissue microenvironment.”

Reviewer 3 Comment #7

- In Figure 6, why was the MEFISTO Factor analysis not performed in the separate patient (autoantibody) groups, since the aim is to find differences between these groups. The pooled analysis may generate inappropriate results. In this respect it would be reassuring to validate well known factors that are tightly correlated to lymphocytic infiltrates (in SSA+ patients eg. CCL19, CXCL13-associated pathways). Such factors previously were shown to yield far more stronger correlations compared what seems to be the case for the loadings of Factor 1 (figure 6C,D). In fact, what are the correlations of the identified loadings with T cells, B cells etc (eg assessed by absolute counts of deconvolution data).

We thank the reviewer for this important comment. We agree that uncovering group-specific molecular features is critical for understanding the heterogeneity of SjD.

Regarding the MEFISTO factor analysis in Figure 6, our decision to perform the analysis on pooled samples across groups (rather than separately for each autoantibody-defined subgroup) was based on a key methodological consideration. When MEFISTO is run independently for each group, the resulting latent factors are derived separately, making it difficult to directly compare factor identities or trajectories between groups. Therefore, we followed the approach described in the original MEFISTO publication (PMID: 35027765), in which pooled data is used to identify shared and group-specific spatial factors across conditions. Using this strategy, we found that Factor 1 was a shared component among all SjD groups (SSA+, CENT+, SSA+CENT+), but not observed in Sicca samples, suggesting that it captures disease-associated programs common to SjD.

We acknowledge that our MEFISTO analysis did not clearly separate SSA+ and CENT+ subgroups. This may reflect limitations of the model when applied to spatial transcriptomics data with modest sample size. Nonetheless, our pathway enrichment analysis in Figure 5 revealed distinct molecular programs between the groups, such as enhanced epithelial-to-mesenchymal transition (EMT)-related signaling in CENT+ SjD and type I IFN response in SSA+ SjD, supporting the presence of meaningful subgroup differences despite the convergence observed in the MEFISTO factor.

In response to the reviewer’s concern regarding the biological validity of Factor 1, we sought to evaluate whether this factor aligns with known immunopathological features of SjD. As shown in **Extended Figure 9**, we examined the correlation between Factor 1 loadings and both (i) cell-type-specific gene expression and (ii) cell-type proportions estimated by spatial deconvolution.

We first assessed the association between Factor 1 and the expression of canonical immune and stromal markers across spots (**Extended Figure 9c**). We observed strong positive correlations between Factor 1 and genes marking T cells (e.g., *CD3E*, *CD4*, *CD8A*), cytotoxic programs (*GZMK*, *GZMB*), B/plasma cells (*CD19*, *SDC1*), myeloid cells (*CD68*), fibroblasts (*THY1*), and chemokines such as *CXCL13* and *CCL19*. These findings suggest that Factor 1 captures a coordinated immune activation and stromal response program within inflamed regions of the salivary gland. Notably, *PRG4*, a marker for a different subset of fibroblast, showed very weak correlation with Factor 1, further supporting the immune-centric specificity of this factor.

To complement this gene-level analysis, we used the Redeconve deconvolution algorithm (Zhou et al., *Nat Commun* 2023, PMID: 38040768), applying our scRNA-seq data as reference from SjD salivary glands to estimate the proportion of T cells and B/plasma cells in each Visium spot. As shown in **Extended Figure 9a,b**, T cell proportions were strongly correlated with Factor 1 scores ($R = 0.58$, $p < 0.01$), whereas B/plasma cell proportions showed minimal correlation ($R = -0.04$, $p < 0.01$). However, given the strong gene-level correlations with *CD19* and *SDC1*, we suspect that the weak correlation from Redeconve pipeline may reflect technical limitations of the deconvolution model rather than the absence of B cell involvement in Factor 1.

a**b****c**
Extended Data Fig. 9: Validation of MEFISTO Factor 1 as a marker of spatial immune activation.

a, Correlation between Factor 1 scores and the proportion of T cells in each Visium spot, as estimated by spatial transcriptomics deconvolution using the Redeconve pipeline with our scRNA-seq data as reference. **b**, Correlation between Factor 1 scores and the proportion of B/plasma cells per spot. **c**, Correlation plots between Factor 1 scores and expression levels of canonical marker genes for various cell types and immune pathways. Genes shown include markers for T cells (*CD3E*, *CD4*, *CD8A*), cytotoxic programs (*GZMK*, *GZMB*), B cells and plasma cells (*CD19*, *SDC1*), myeloid cells (*CD68*), fibroblasts (*PRG4*, *THY1*), and key immune chemokines (*CXCL13*, *CCL19*).

Taken together, these analyses support the conclusion that Factor 1 reflects immune infiltration within SjD tissues, and that its spatial variation corresponds well with known pathogenic features.

While we acknowledge that this approach is a proxy and lacks single-cell precision, it offers a useful approximation of lymphocyte-rich areas in the tissue. Future application of subcellular-resolution platforms such as Xenium may help disentangle cell-type-specific spatial programs with greater clarity.

We have included these new results in Extended Data Figure 9, and we have updated the manuscript text accordingly. We thank the reviewer again for this important suggestion, which prompted us to more comprehensively validate the biological relevance of the inferred latent factors.

Results, Line 515, Page 14:

“To further characterize the biological relevance of Factor-1, we investigated its association with spatial immune cell infiltration and gene expression. Using the Redeconve pipeline⁴², with our scRNA-seq dataset from salivary glands as a reference, we deconvolved each Visium spot and estimated the proportions of T cells and B/plasma cells. Factor-1 scores were strongly correlated with T cell proportion across spatial spots ($R = 0.58$, $p < 0.01$; **Extended Data Fig. 9a**), whereas only a weak inverse correlation was observed with B/plasma cell proportion ($R = -0.04$, $p < 0.01$; **Extended Data Fig. 9b**). This discrepancy may reflect limitations in deconvolution sensitivity for B lineage cells rather than a true absence of association. Indeed, Factor-1 values exhibited strong positive correlations with markers of B cells and plasma cells (*CD19*, *SDC1*), as well as T cells (*CD3E*, *CD4*, *CD8A*), cytotoxicity (*GZMK*, *GZMB*), and myeloid cells (*CD68*), as well as the chemokines *CXCL13* and *CCL19*, which are known to be upregulated in lymphocyte-rich regions (**Extended Data Fig. 9c**). In contrast, *PRG4*, the marker of a different subset of fibroblast, showed minimal correlation, underscoring the immune-specific nature of Factor-1. These findings support the interpretation of Factor-1 as a latent axis of spatial immune activation that integrates signals from multiple immune lineages and stromal components.”

Minor Comments:

Reviewer 3 Comment #8

- Several observations come with some uncertainty because of differences in group size. Eg. In Figure 2H in SSA+ patients (n=8) GZMB+GNLY+CD8+ show an enrichment of the EBV pathway (likely type I IFN), but not in the SSA+CENT+ patients (n=4) nor in CENT+ patients (n=4). Is this not just a power issue?

We thank the reviewer for raising this important point. We agree that differences in sample size across groups might influence the power to detect pathway enrichments, particularly in smaller cohorts such as the CENT+ and SSA+CENT+ groups.

To address this concern, we expanded the CENT+ Sjd group from 4 to 7 samples in our scRNA-seq dataset and repeated the differential gene expression analyses for CD8+ T cells. Notably, *MX1*, a canonical interferon-stimulated gene (ISG), was significantly upregulated in both SSA+ and SSA+CENT+ patients compared to Sicca controls, but not in the expanded CENT+ group (Fig. 2g and Supplementary Table 2). This suggests that, while limited power may play a role, the type I interferon response appears to be more prominent in SSA+ patients than in CENT+ patients in our dataset.

Fig. 2g: Differentially expressed gene (DEG) analysis in CD4⁺ and CD8⁺ T cells across Sjd subgroups compared to Sicca. Volcano plots show log₂ fold changes (log₂FC) (x-axis) and -log₁₀ adjusted p-values (y-axis) for DEGs between each Sjd subgroup and Sicca controls in CD4⁺ (left three panels) and CD8⁺ T cells (right three panels). Upregulated genes (red) and downregulated genes (blue) are highlighted with selected key genes annotated. Genes not reaching significance are shown in grey.

This interpretation is further supported by our spatial transcriptomics (Visium) data, where type I IFN-related pathways were specifically enriched in SSA+ samples, whereas CENT+ patients exhibited alternative pathway signatures such as epithelial-to-mesenchymal transition (EMT) (Fig. 5h). Furthermore, this pattern aligns with previous studies reporting that SSA+ Sjd exhibits a stronger IFN gene signature than SSA- subgroups, including those likely enriched for CENT+ patients (PMID: 34112769).

Fig. 5h: Gene Set Enrichment Analysis (GSEA) comparing autoantibody status using Visium data.

Only selected pathways with significance (FDR<0.05) in spot cluster 4 are shown. The size of the dots represents the set size of genes involved, and the color intensity indicates the normalized enrichment score (NES).

We cannot completely rule out power limitations, and this does not indicate the absence of IFN activation in CENT+ patients—indeed, DEG analysis comparing CENT+ to Sicca revealed increased expression of IFN-related genes such as *ISG15* (Supplementary Data 4). Rather, these analyses suggest that the magnitude of IFN pathway activation is relatively higher in SSA+ patients compared to CENT+ patients. We thank the reviewer for prompting a careful consideration of this issue.

- Harmonization of single-cell data (Mat. & Meth., line 494): The manuscript mentions a harmonization step for single-cell data. Since the single-cell analysis appears to have been conducted immediately after tissue dissociation, with samples potentially processed on different days, it is unclear what specific batch effects are being addressed. Additional details on the rationale and implementation of this harmonization would be helpful.

We thank the reviewer for this important question regarding the rationale and implementation of batch harmonization in our single-cell analysis.

Following technical noise removal using CellBender, we applied Harmony to correct for potential batch effects introduced during sample processing. Although the tissues were processed immediately after dissociation, samples were handled and sequenced on different days, and there were differences in PCR amplification cycles across runs, which can introduce technical variability in scRNA-seq data. Harmony was used to mitigate these effects by aligning cells across batches based on processing date and sequencing metrics.

In addition, we used Harmony to adjust for donor-specific effects by incorporating sample ID as a covariate, which allowed us to minimize variation arising from inter-individual differences not directly related to disease status. Furthermore, during downstream statistical modeling (e.g., differential abundance testing), we also adjusted for age and sex to further control for potential confounders.

Together, these steps were taken to minimize the influence of technical and biological variability unrelated to disease phenotype, thereby allowing us to more accurately identify molecular and cellular differences associated with autoantibody-defined subgroups of SjD. We have clarified this rationale in the revised Materials and Methods section.

- Please use new nomenclature for Sjogren syndrome -> Sjogren's disease

We appreciate and agree with the reviewer's comment in terms of this terminological update. We have revised the manuscript accordingly by replacing all instances of "Sjögren's syndrome (SjS)" with "Sjögren's disease (SjD)" throughout the text.

Reviewers' comments:

Reviewer 4 (Remarks to the Author):

We appreciate your taking the time to co-review our manuscript and for the collaborative approach you have taken with your colleague.

Reviewers' comments:**Reviewer 2 Comment #1**

- The authors made sincere effort in trying to address the reviewers' comments to the best of their capability. The manuscript remains, in some cases, still hampered by the same problems which were identified previously. Altogether, however, most of the responses are satisfactory and the manuscript has substantially improved.

We sincerely thank the reviewer for their thoughtful evaluation and for acknowledging the improvements made in the revised manuscript. We appreciate the recognition of our efforts to address the prior comments to the best of our capability.

We also take note of the reviewer's remark that some issues may still remain. We remain committed to clarity and rigor and have further revised the manuscript to improve these aspects wherever possible in this final version.

We are grateful for the constructive feedback, which has been invaluable in strengthening the quality of the manuscript.

Reviewer 3 Comment #1

- The authors have given detailed answers to all the raised points. I think the added analyses and clarifications have significantly improved the manuscript. There is one point the authors should address: According to the ACR/EULAR criteria to classify as a SjD the patients should have objectified dryness symptoms and either an LFS of 1 or higher or SSA positivity. For the CENT+ group this is not clear. Only 1 is SSA+ and the LFS median values don't give insight. Maybe give means or indicate which percentage has LFS 1 or higher. If the patients should not qualify, consider them CENT sicca. Were Schirmer and UWS quantified? Please add these to the table

We thank the reviewer for this important comment. To address this point, we have now added the proportion of patients with $LFS \geq 1$ and those with Schirmer positivity for each group in Supplementary Table 1. Unfortunately, our hospital does not routinely measure unstimulated whole saliva flow rate, so we did not have any data available. This additional information clarifies the objective dryness status of the CENT+ SjD group and other subgroups.

Reviewer 4 (Remarks to the Author):

We thank the reviewer for their time and thoughtful evaluation of our manuscript.